# Role of eyewall and rainband eddy forcing in tropical cyclone intensification

Ping. Zhu[1], Bryce Tyner[1], Jun A. Zhang[3,4], Eric Aligo[2], Sundararaman Gopalakrishnan[3], Frank D. Marks[3], Avichal Mehra[2], Vijay Tallapragada[2],

[1]Department of Earth and Environment, Extreme Event Institute, Florida International University, Miami, FL 33199, US
[2]Environmental Modeling Center, NCEP, NOAA, College Park, MD 20740, US
[3]Hurricane Research Division, AOML, NOAA, Miami, FL 33149, US
[4]Cooperative Institute for Marine and Atmospheric Studies, University of Miami, Miami, FL 33149, US

*Correspondence to*: Ping Zhu (zhup@fiu.edu)

**Abstract.** While turbulence is commonly regarded as a flow feature pertaining to the planetary boundary layer (PBL), intense turbulent mixing generated by cloud processes also exists above the PBL in the eyewall and rainbands of a tropical cyclone (TC). The in-cloud turbulence above the PBL is intimately involved in the development of convective elements in the eyewall and rainbands and consists of a part of asymmetric eddy forcing for the evolution of the primary and secondary circulations of a TC. In this study, we show that the Hurricane Weather Research & forecasting (HWRF) model, one of the operational models used for TC prediction, is unable to generate appropriate sub-grid-scale (SGS) eddy forcing above the PBL due to lack of consideration of intense turbulent mixing generated by the eyewall and rainband clouds. Incorporating an in-cloud turbulent mixing parameterization in the vertical turbulent mixing scheme notably improves HWRF's skills on predicting rapid changes in intensity for several past major hurricanes. While the analyses show that the SGS eddy forcing above the PBL is only about one-fifth of the model-resolved eddy forcing, the simulated TC vortex inner-core structure, secondary overturning circulation, and the model-resolved eddy forcing exhibit a substantial dependence on the parameterized SGS eddy processes. The results highlight the importance of eyewall/rainband SGS eddy forcing to numerical prediction of TC intensification, including rapid intensification at the current resolution of operational models.

## 1. Introduction

Producing timely and accurate intensity forecasts of tropical cyclones (TCs) continues to be one of the most difficult challenges in numerical weather prediction. The difficulty stems from the fact that TC intensification is not only modulated by environmental conditions, such as large-scale wind shear and underlying sea surface temperature (SST), but also largely depends on TC internal dynamics that involve complicated interactions of physical processes spanning over a spectrum of scales (Marks and Shay 1998). Since numerical models use discretized grids to simulate the continuous atmosphere, the processes with scales smaller than model grid spacing, known as sub-grid scale (SGS) processes, cannot be resolved by

models. Because of the high nonlinearity of the atmospheric system, the SGS processes result in new high-order terms in the grid-box-mean governing equations of the atmosphere. These new high-order terms cause the otherwise closed system no longer to be closed. To close the system, additional equations that govern high-order terms need to be derived. This is the notorious closure problem of any turbulent fluid system. In practice, the high-order terms are determined parametrically in terms of model-resolved grid-box mean variables, known as turbulent mixing parameterization.

TC intensification associated with internal dynamics including SGS processes may be better approached in a cylindrical coordinate with its origin set at the center of a TC vortex. The governing equation for the azimuthal-mean model-resolved tangential velocity of a TC may be written as:

$$\frac{\partial \bar{\tilde{v}}}{\partial t} + \bar{\tilde{u}}\frac{\partial \bar{\tilde{v}}}{\partial r} + \bar{\tilde{w}}\frac{\partial \bar{\tilde{v}}}{\partial z} = -\bar{\tilde{u}}\left(f + \frac{\bar{\tilde{v}}}{r}\right) + F_\lambda + F_{sgs\_\lambda}, \qquad F_\lambda = -\overline{\tilde{u}'\frac{\partial \tilde{v}'}{\partial r}} - \overline{\tilde{v}'\frac{\partial \tilde{v}'}{r\partial\lambda}} - \overline{\tilde{w}'\frac{\partial \tilde{v}'}{\partial z}} - \frac{\overline{\tilde{u}'\tilde{v}'}}{r}, \tag{1}$$

where $r$, $\lambda$, and $z$ represent the radial, azimuthal, and vertical coordinate axes; $\tilde{u}$, $\tilde{v}$, and $\tilde{w}$ are the model-resolved radial, tangential, and vertical wind components, respectively; $f$ is Coriolis parameter. Overbar and prime indicate the azimuthal-mean and the perturbation away from the azimuthal-mean; $F_\lambda$ is the azimuthal-mean tangential eddy correlation term resulting from the model-resolved asymmetric eddy processes; and $F_{sgs\_\lambda}$ is the azimuthal-mean tangential SGS tendency resulting from the parameterized SGS eddy processes (or turbulence). In the region where friction is appreciable, eddy forcing $F_\lambda + F_{sgs\_\lambda}$ is negative definite (Montgomery and Smith 2014), meaning that it tends to slow down the motion. Defining the azimuthal-mean model-resolved absolute angular momentum per unit mass as $\bar{\tilde{M}} = r\bar{\tilde{v}} + \frac{1}{2}fr^2$, it is easy to show that the azimuthal-mean tangential wind budget equation, Eq. (1), becomes:

$$\frac{D\bar{\tilde{M}}}{Dt} = \frac{\partial \bar{\tilde{M}}}{\partial t} + \bar{\tilde{u}}\frac{\partial \bar{\tilde{M}}}{\partial r} + \bar{\tilde{w}}\frac{\partial \bar{\tilde{M}}}{\partial z} = r(F_\lambda + F_{sgs\_\lambda}), \tag{2}$$

where $\frac{D\bar{\tilde{M}}}{Dt} = \frac{\partial \bar{\tilde{M}}}{\partial t} + \bar{\tilde{u}}\frac{\partial \bar{\tilde{M}}}{\partial r} + \bar{\tilde{w}}\frac{\partial \bar{\tilde{M}}}{\partial z}$ is the material derivative following air particles along the model-resolved axisymmetric flow and $r(F_\lambda + F_{sgs\_\lambda})$ is the torque per unit mass acting on air parcels resulting from the model-resolved and SGS eddy forcing. For $F_\lambda + F_{sgs\_\lambda} = 0$, $\bar{\tilde{M}}$ is materially conserved. As air parcels move radially inward (decrease of $r$), they must spin up in order to conserve their absolute angular momentum. Conversely, air parcels must spin down as they move radially outward. This provides an essential mechanism for the spin-up process of a vortex free of forcing in an inviscid flow. For $F_\lambda + F_{sgs\_\lambda} \neq 0$, the asymmetric eddy processes provide an important forcing for the evolution of the primary circulation of a TC vortex as indicated by Eq. (1) or Eq. (2).

The asymmetric eddies that produce tangential eddy forcing for driving the mean vortex circulation cover a spectrum of scales from mesovortices, mesoscale convective plumes, down to small scale turbulence. The advanced three-dimensional

(3D) rotating convective updraft paradigm (Montgomery and Smith 2014) recognized the importance of asymmetries, such as hot towers, to TC intensification. Persing et al. (2013) compared the TC intensification rate in a 3D full-physics model with that in an axisymmetric model. Their results show that the 3D eddy processes associated with vortical plumes can assist the intensification process by contributing to the azimuthally averaged heating rate, to the radial contraction of the maximum tangential velocity, and to the vertical extension of tangential winds through the depth of the troposphere. Since mesoscale convective plumes can be explicitly resolved by high resolution regional models, the 3D full-physics simulations provide a means to elucidate the role of the model-resolved eddy forcing in TC intensification.

Small-scale turbulence including large energy-containing eddies (e.g., sub-kilometer convective elements and roll vortices) cannot be resolved by 3D full-physics regional models. The parameterized turbulent mixing results in the SGS eddy forcing (e.g., $F_{sgs\_\lambda}$) for the evolution of a TC vortex in numerical simulations. Since turbulence is a basic flow feature pertaining to the planetary boundary layer (PBL), SGS eddy forcing is commonly considered to be important only in the PBL. The importance of PBL turbulence to TC evolution has been recognized for a long time. Both the Conditional Instability of the Second Kind (CISK) and cooperative-intensification mechanism (Ooyama 1982), the two early theories for TC intensification, recognized the role of the PBL in converging moisture to sustain deep convection of a TC. Charney and Eliassen (1964) stated, "Friction performs a dual role; it acts to dissipate kinetic energy, but because of the frictional convergence in the moist surface boundary layer, it acts also to supply latent heat energy to the system". Later it was Emanuel's evaporation-wind feedback mechanism (Emanuel 2003) that first articulated the critical role of air-sea interaction in generating a positive feedback between the near-surface wind speed and the rate of evaporation from the underlying ocean during the intensification process. However, in all these theories plus the 3D rotating convective updraft paradigm (Montgomery and Smith 2014), the PBL was treated as a shallow turbulent layer adjacent to Earth's surface with a depth typically less than 1 km. By doing so, they implicitly adopted the basic assumptions of the classic PBL theory: (1) turbulent mixing is responsible for the vertical transport of momentum, heat, and moisture; (2) vertical turbulent transport becomes negligible above the PBL; and (3) the mean vertical velocity $\overline{w}$ in the PBL is negligible compared with the vertical velocity fluctuations $w'$ (Stull 1988).

However, turbulence and the resultant turbulent transport and turbulent kinetic energy (TKE) cannot always be neglected above the PBL. Intense turbulent mixing within the deep convective clouds has been widely observed by aircraft, Doppler radar/lidar, and other advanced remote sensing instruments (e.g., LeMone and Zipser 1980; Marks et al. 2008; Hogan et al. 2009; Giangrande et al. 2013). In particular, using the TKE derived from the airborne radar data collected in Hurricane Rita (2005), Lorsolo et al. (2010) showed that large TKE exists above the PBL in the eyewall and rainbands. Figure 1 shows the composite of TKE derived using Lorsolo et al. (2010)'s method based on the airborne radar observations from 116 radial legs of P3 flights in the 2003-2010 hurricanes seasons. It clearly demonstrates that the intense turbulence exists above the PBL in the conventional definition all the way up to over 10 km in the eyewall.

Realizing the deep convective nature of TCs, Smith et al. (2008) and Smith and Montgomery (2010) warned that the conventional PBL theory may become invalid in the TC inner-core region as the low-level radial inflow ascends swiftly within the eyewall. In fact, the problem of applying conventional PBL theory to a deep convective regime had been recognized early in the 1970s and 1980s. Deardorff (1972) noted, "*The definition of PBL has not included the region of turbulence within towering cumuli but only the average height of surface induced turbulent fluxes outside of such clouds*". Moss and Rosenthal (1975) added, "*The method (of defining the PBL) contains several elements that may or may not be applicable under hurricane conditions*". Shapiro (1983) wrote, "*As the radius of maximum tangential wind is approached, the boundary layer itself becomes ill defined, as air is pulled up into the active convection*". Stull (1988) also acknowledged that "*the conventional definition of PBL is not applicable to the intertropical convergence zone (ITCZ), where the air ascends into deep convective clouds*".

The problem here, however, is not all about how to redefine PBL to encompass all the scenarios including the deep convective regime. This is because the concept of PBL always applies as to the layer adjacent to the surface that is directly affected by the surface processes. From the perspective of TC intensification, the real questions that need to be answered are: (1) Is the intense turbulent mixing above the PBL in the eyewall and rainbands generated by cloud processes important to TC intensification? And (2) how does the parameterized in-cloud eddy processes in the eyewall and rainbands affect model resolved eddy forcing and TC inner-core structure? The answer to the first question is apparent as in-cloud turbulence results in a component of direct eddy forcing for the mean circulation of a vortex according to Eq. (1) or Eq. (2). The complication is that the sign of eddy forcing, $F_\lambda + F_{sgs\_\lambda}$, above the PBL is indefinite depending on the details of eddy processes. Persing et al. (2013) showed that the resolved 3D momentum fluxes above the PBL exhibit counter-gradient characteristics during a key spin-up period, and more generally are not solely diffusive. Thus, for $F_\lambda + F_{sgs\_\lambda} > 0$, it provides a mechanism for spinning up a vortex. The second question is important since in numerical simulations the asymmetric eddies with a continuous spectrum are artificially split into the model-resolved and parameterized components because of the discretized model grids. The two split parts of eddy forcing are not independent but interact with each other depending on the model resolution. To date, little work has been done to examine the sensitivity of model-resolved eddy forcing and TC structure to the parameterized SGS eddy processes above the PBL generated by the eyewall/rainband clouds. This issue will be investigated in this study.

In addition to the direct tangential eddy forcing $F_\lambda + F_{sgs\_\lambda}$ to the primary circulation of a TC vortex, the secondary overturning circulation induced by friction and diabatic heating also plays an important role in TC intensification. The azimuthal-mean governing equations for model-resolved radial and vertical velocities of the overturning circulation may be written as:

$$\frac{D\bar{\tilde{u}}}{Dt} - C = -\frac{1}{\bar{\tilde{\rho}}}\frac{\partial \bar{\tilde{p}}}{\partial r} + F_r + F_{sgs\_r}, \quad C = \frac{\bar{\tilde{v}}^2}{r} + f\bar{\tilde{v}}, \quad F_r = -\overline{\tilde{u}'\frac{\partial \tilde{u}'}{\partial r}} - \overline{\tilde{v}'\frac{\partial \tilde{u}'}{r\partial \lambda}} - \overline{\tilde{w}'\frac{\partial \tilde{u}'}{\partial z}} - \overline{\frac{\tilde{v}'^2}{r}}, \tag{3}$$

$$\frac{D\bar{\tilde{w}}}{Dt} = -\frac{1}{\bar{\tilde{\rho}}}\frac{\partial \bar{\tilde{p}}}{\partial z} - g + F_w + F_{sgs\_w}, \qquad F_w = -\overline{\tilde{u}'\frac{\partial \tilde{w}'}{\partial r}} - \overline{\tilde{v}'\frac{\partial \tilde{w}'}{r\partial \lambda}} - \overline{\tilde{w}'\frac{\partial \tilde{w}'}{\partial z}}, \tag{4}$$

where $F_r + F_{sgs\_r}$ and $F_w + F_{sgs\_w}$ are the model-resolved and SGS eddy forcing terms in the radial and vertical direction; $\bar{\tilde{p}}$ and $\bar{\tilde{\rho}}$ are the azimuthal-mean model-resolved pressure and air density, respectively; and $g$ is the gravitational acceleration.

In the classic TC studies (e.g., Ooyama 1969 and Emanuel 2003), TC vortices were assumed to follow the gradient wind balance and hydrostatic balance where the accelerations of radial and vertical velocities ($\frac{D\bar{\tilde{u}}}{Dt}, \frac{D\bar{\tilde{w}}}{Dt}$) and the radial and vertical eddy forcing ($F_r + F_{sgs\_r}, F_w + F_{sgs\_w}$) in Eq. (3) and Eq. (4) are neglected. Shapiro and Willoughby (1982); Smith et al. (2005), and Bui et al. (2009) showed that in such a balanced framework the secondary overturning circulation of a TC vortex can be analytically described by an elliptical partial differential equation known as Sawyer-Eliassen equation (SEE). Using this diagnostic tool, Shapiro and Willoughby (1982) examined the acceleration of tangential wind in response to local sources of heating and momentum. Later, Smith et al. (2009) showed that the convergence of absolute angular momentum within the PBL associated with the development of super-gradient wind speeds can provide a spin-up mechanism for the mean tangential circulation of a vortex. Therefore, intensification theories built upon gradient wind balance and hydrostatic balance may lack the ability to explain the rapid intensity changes driven by the internal dynamics when radial or vertical eddy forcing becomes important. Numerical models built upon primitive equations presumably have the ability to capture the eddy forcing associated with convection and PBL turbulence. Advances in computer technology nowadays have reduced model horizontal grid spacing of operational models down to 1-2 km. While higher resolution models allow dynamic eddy forcing ($F_\lambda, F_r, F_w$) and thermodynamic eddy forcing for heat and moisture ($F_\theta, F_q$) to be better resolved, it remains to be poorly understood as to what governs the sign, magnitude, and radius-height distribution of eddy forcings above the PBL. Leaving aside the question if high resolution numerical models can generate robust model-resolved eddy forcing, a source of uncertainty in intensity forecast arises from the parametric determination of SGS eddy processes.

In numerical models, the SGS forcings ($F_{sgs\_\lambda}, F_{sgs\_r}, F_{sgs\_w}, F_{sgs\_\theta}$, and $F_{sgs\_q}$) are determined by the turbulent mixing scheme. Current effort mainly focuses on the improvement of parameterization of turbulent mixing within the PBL. The importance of eyewall/rainband SGS eddy forcing above the PBL to TC intensification has been largely overlooked in the past for a few reasons. First, the critical role of radial inflow, PBL processes, and surface latent heating in maintaining and intensifying a TC vortex has overshadowed the importance of the SGS forcing aloft associated with eyewall/rainband convection. Second, unlike turbulence in the PBL, which has a solid theory built upon observations, turbulence aloft in deep convection is difficult to access. Lack of observations largely limits our understanding of the in-cloud turbulent mixing processes and the resultant SGS eddy forcing to the momentum and heat budgets of a TC. Third, for deep convection, the

focus is on the cumulus parameterization. Cumulus schemes (e.g., Arakawa and Schubert 1974; Betts and Miller 1993) were originally designed to remove the convective instability generated by the large-scale flow and alter the thermodynamic structure of the environment based on the parameterized convective fluxes and precipitation. It is commonly assumed and widely accepted that the coherent up-/down-drafts take the central role in establishing the equilibrium between the generation of moist convective instability by the environmental processes and the stabilization of environment by cumulus convection (Arakawa and Schubert 1974; Wu and Arakawa 2014; Zhu 2015). The effect of small scale turbulence is negligible in this perspective although some later developed more advanced cumulus schemes do consider the effects of turbulent mixing in schemes (e.g., Guo et al. 2015). Finally, almost all turbulent mixing schemes used today for TC prediction were originally developed to represent turbulent processes within the PBL in fair-weather conditions in which the turbulent PBL is cleanly separated from the free atmosphere above by a capping inversion. Often in these schemes, a simple method based on the bulk Richardson number is adopted to account for the free atmosphere turbulence if there is any (e.g., Hong and Pan, 1996). These schemes lack the ability to represent the in-cloud turbulence in the eyewall and rainbands generated by the cloud processes. Thus, the contribution of in-cloud turbulence above the PBL to eddy forcing in TC intensification, and the sensitivity of resolved eddy forcing and vortex inner-core structure to the parameterization of in-cloud turbulence are largely unknown.

TC intensification is a complicated process that is affected by a number of environmental factors, such as wind shear and SST. Emanuel et al. (2004) examined the sensitivity of storm intensity simulated by a coupled axisymmetric model known as the Coupled Hurricane Intensity Prediction System (CHIPS) to vortex initialization and various environmental factors. Their results showed that the simulated storm intensity is most sensitive to wind shear. Recently, Vigh et al. (2018) confirmed Emanuel et al. (2004)'s results and showed that very rapid intensification (VRI, ~30 kt in 12 h) and extreme rapid intensification (ERI, ~40 kt in 12 h) can be well captured by CHIPS with the setting of zero wind shear. While environmental conditions appear to be critical to TC intensification, they will not be discussed in this study, rather, we focused on how does eddy forcing resulting from both resolved and parameterized asymmetric eddy process modulate TC intensification under certain environment conditions. In particular, using numerical simulations by the Hurricane Weather Research and Forecast (HWRF) modeling system, one of the operational models used for TC prediction at the Environmental Modeling Center (EMC), NOAA, we investigate the role of eyewall/rainband eddy forcing in governing TC intensity change. We demonstrate the sensitivity of intensification process to parameterization of eyewall/rainband in-cloud turbulent mixing above the PBL in numerical simulations of TCs. This paper is organized as follows. In section 2, we show problems associated with the turbulent mixing scheme used in the operational HWRF in representing eyewall/rainband in-cloud turbulence and discuss methods of how to incorporate the parameterization of in-cloud turbulence in the PBL scheme used in HWRF. The simulation results by the HWRF with the operational setting and the modified PBL scheme that includes an in-cloud turbulent mixing parameterization are presented in section3 followed by a summary in section 4.

## 2. HWRF PBL scheme and in-cloud turbulent mixing parameterization

The numerical model used in this study is the operational HWRF version 3.8a. It consists of triple-nested domains on an E-grid. The grid-spacing of the three domains is $0.135^0$, $0.045^0$, and $0.015^0$ degree, corresponding approximately to 18 km, 6 km, and 2 km, respectively. There are 61 levels in the vertical. The details of HWRFv3.8a release can be accessed at https://dtcenter.org/HurrWRF/users/docs/index.php. Since this study focuses on the role of internal eyewall/rainband eddy forcing in TC intensification, to avoid the complication from the interactive underlying ocean, all simulations presented in this paper were performed by the uncoupled atmospheric model of HWRF. The initial and boundary conditions for the real-case TC simulations were supplied by the Global Forecast System (GFS) data.

As discussed earlier, in numerical models the SGS eddy forcing is determined by the turbulent mixing scheme. Since large energy-containing turbulent eddies are not resolved by 2-km resolution grids, to appropriately parameterize the anisotropic SGS eddy processes, like other state-of-the-art regional models, the operational HWRF treats horizontal and vertical turbulent mixing separately. The horizontal SGS mixing is handled by a revised two-dimensional (2D) Smagorinsky diffusion model (Zhang et al. 2018) that is built within the model dynamic core. The vertical SGS mixing, on the other hand, is handled by a separate physics module known as the PBL scheme. It is a 1D vertical turbulent mixing scheme, which was formulated based on the one originally proposed by Hong and Pan (1996). Bryan and Rotunno (2009) and Bryan (2012) investigated the sensitivity of TC evolution to horizontal eddy diffusivity by adjusting the mixing length. Recently, Zhang et al. (2018) evaluated the impact of horizontal diffusion parameterization on TC prediction by HWRF. In this study, we only focus on the vertical turbulent mixing parameterization. Horizontal diffusion was not touched.

The HWRF PBL scheme is a typical K-closure (or first-order-closure) turbulent mixing scheme. Although there have been modifications to the scheme throughout the years, the basic formulae used to determine eddy exchange coefficients are kept the same as those originally proposed by Hong and Pan (1996). In this scheme, the eddy exchange coefficients are determined separately based on the diagnosed PBL height. Within the PBL, the momentum eddy viscosity is calculated as:

$$K_m = \kappa \frac{u_*}{\phi_m} \alpha z (1 - \frac{z}{h})^2, \tag{5}$$

where $\kappa$ is the von Karman constant, $u_*$ is the friction velocity, $z$ is the height above the ground surface, $\phi_m$ is the surface layer stability function obtained by Businger et al. (1971), and $h$ is the diagnosed PBL height calculated iteratively based on the bulk Richardson number over the PBL depth and the buoyancy of surface-driven thermals. Although there are many sophisticated methods to parameterize SGS turbulent mixing, such as TKE closure, high-order closure, nonlocal mixing, and schemes formulated using variables conserved for moist reversible adiabatic processes, the K-closure scheme is arguably the best choice for operational models at the current stage as it requires the least computational resource. However, Eq. (5) was originally formulated to account for PBL turbulent mixing in non-TC conditions (Troen and Mahrt 1986; Holtslag et al. 1990; Holtslag and Boville 1993). Observations from multiple TCs by Zhang et al. (2011) showed that Eq. (5) substantially

overestimates the eddy viscosity in the PBL. In light of this finding, Gopalakrishnan et al. (2013) introduced a coefficient $\alpha$ ($0< \alpha <1$) in Eq. (5) to reduce eddy viscosity in TC simulations. This tuning of eddy viscosity via $\alpha$ now has been adopted in the operational HWRF. Above the diagnosed PBL height, the momentum eddy viscosity is calculated as:

$$K_m = l^2 f_m(Ri_g) \sqrt{\left|\frac{\partial \widetilde{u}}{\partial z}\right|^2 + \left|\frac{\partial \widetilde{v}}{\partial z}\right|^2}, \tag{6}$$

where $l$ is the mixing length, $f_m(Ri_g)$ is the stability function of gradient Richardson number, $Ri_g = \frac{g}{\theta_0}\frac{\partial \widetilde{\theta_v}}{\partial x} / (\left|\frac{\partial \widetilde{u}}{\partial z}\right|^2 + \left|\frac{\partial \widetilde{v}}{\partial z}\right|^2)$,

and $\sqrt{\left|\frac{\partial \widetilde{u}}{\partial z}\right|^2 + \left|\frac{\partial \widetilde{v}}{\partial z}\right|^2}$ is the vertical wind shear. This is a method that was originally proposed to account for the free-atmosphere diffusion. Once $K_m$ is determined, the eddy viscosity for heat and moisture is calculated by $K_{t,q} = K_m P_r^{-1}$, where $P_r$ is the Prandtl number.

For fair-weather conditions, the parameterization formulated by Eq. (5) and Eq. (6) provides a practical way to appropriately parameterize the SGS turbulent mixing within and above the PBL since the turbulent layer resulting from the surface processes is often cleanly separated from the free atmosphere by a capping inversion. The mid-point of the inversion zone (or entrainment zone) is naturally defined as the PBL height (Stull 1988). In a TC environment, however, turbulence is no longer solely generated by the shear production and buoyancy production associated with the surface processes; it can also be generated by cloud processes aloft due to cloud radiative cooling, evaporative cooling, and inhomogeneous diabatic heating and cooling in the clouds. Thus, although the concept of PBL is still applicable, it becomes ambiguous from the turbulent mixing perspective. In many TC studies, the PBL is defined either as the turbulent layer that is directly affected by the surface processes or as the inflow layer of the secondary circulation. But in either case, the so-defined PBL height is by no means a physical interface that separates the turbulence generated by surface processes and by cloud processes. This is particularly true in the eyewall and rainbands of a TC, where intense turbulence can extend from the surface all the way up to the upper troposphere, as was illustrated in Fig. 1. Thus, from the nature of turbulent mixing, an artificial separation of turbulence using a diagnosed "PBL" height is not a physically sound method to parameterize the internally connected SGS turbulent mixing in the eyewall or any deep convective areas in a TC. Moreover, an artificial separation of the PBL from the free atmosphere above may create an unrealistic discontinuity in the vertical profile of eddy viscosity in this method. Following Eq. (5), as height $z$ approaches the diagnosed "PBL" height $h$, eddy viscosity $K_m$ becomes zero to result in zero turbulent mixing at a certain model grid level if the diagnosed PBL height falls exactly at this level. Above the diagnosed PBL, the turbulent mixing jumps to whatever value estimated by Eq. (6). This singular point in the vertical profile of eddy exchange coefficient could cause problems in representing turbulent mixing in the eyewall and rainbands.

We carefully examined the eddy exchange coefficients in multiple TC simulations by the operational HWRF and found that the default PBL scheme is unable to generate intense turbulent mixing in the eyewall and rainbands. As an example, Figure 2 shows the horizontal distribution of the HWRF simulated eddy exchange coefficients for momentum ($k_m$) at different

altitudes and the corresponding azimuthal mean of $k_m$ on the radius-height plane of Hurricane Jimena (2015) at an arbitrary time before the storm reached its maximum intensity. Within the PBL, the magnitude and horizontal spatial distribution reflects well the strong turbulent mixing in the eyewall and rainbands, but above the PBL, the HWRF generated eddy exchange coefficients are virtually zero. This result suggests that the PBL scheme used in the operational HWRF fails to capture the intense turbulent mixing above the PBL in the deep convective eyewall and rainbands. This is not a surprise since Eq. (6) was originally developed to parameterize clear-sky free-atmosphere diffusion and is incapable of representing the intense turbulent mixing generated by cloud processes. We hypothesize that the lack of appropriate SGS eddy forcing associated with deep convection above the PBL in the eyewall and rainbands is one of the culprits for the intensity forecast failure in many cases of HWRF forecasts.

To better understand the characteristics of intense turbulent mixing in eyewall clouds, we performed a series of large eddy simulations (LESs) of Hurricane Isabel (2003) in a hindcasting mode using WRF model with the Advanced Research WRF (ARW) dynamic core. The detailed procedure of configuring a WRF-LES for TC simulations can be found in Zhu (2008a and 2008b) and Zhu et al. (2015). The approach of our LES study is similar to that of Bryan et al. (2003) and Green and Zhang (2015) in that the model horizontal grid resolution falls in the Kolmogorov inertial subrange and a 3D SGS model built within the model dynamic solver is used to treat the horizontal and vertical mixing induced by the presumably isotropic SGS eddies. Since eddies with scales smaller than inertial subrange contain much less energy and are less flow-dependent than large energy-containing eddies, the LES methodology is commonly thought to be insensitive to formulaic details and arbitrary parameters of the SGS model, and thus, the turbulent flow generated by LESs are often used as a proxy for reality and a basis for understanding turbulent flow and guiding theories when direct observations are difficult to obtain. In the past, LESs were mainly used to elucidate problems associated with the turbulent processes within the PBL. Here we use this approach to better understand the turbulent processes in the eyewall.

In this LES study, the innermost domain of the WRF-LES covered the entire eyewall of Isabel (2003) with a horizontal grid-spacing of 100 m. 75 levels were configured in the vertical. The simulation was initialized and forced by the NCEP FNL analyses and run for 8 hours (from 00:00 to 8:00 UTC 12 September 2003). The details and results of this Giga WRF-LES is reported in Li et al. (2019). Figure 3 shows the instantons surface (10-m) wind speeds of Isabel (2003) at the 8[th] simulation hour from one of the LESs that uses the 3D nonlinear backscatter and anisotropy (NBA) SGS model (Kosović, 1997). Eyewall disturbances with scales of a few kilometers or smaller are clearly shown in the wind fields. These kilometer-scale or sub-kilometer-scale eddies have been also reported in previous LES studies of TCs. For example, Rotunno et al. (2009) found that these 'vigorous small-scale eddies' are the dominant features in the eyewall in their LES run at the resolution of 62 m. Green and Zhang (2015) showed such disturbances existing in all of their LES runs with the 3D NAB SGS model including the simulation at 333-m resolution.

While using LES to simulate TC is promising, evaluation of the fidelity of the simulated TC vortex and the associated fine-scale structures resolved by LES is a challenge. In the absence of decisive observational measurements, the principal method of evaluating LES has been through sensitivity studies of individual LESs with different SGS models or inter-comparisons among different LESs. The logic is that the robustness of the simulations testifies to its fidelity. Such sensitivity tests and inter-comparison studies in the past have shed favorable light on the LES approach in general in many meteorological applications (e.g., Stevens et al. 2005; Moeng et al. 1996), but they also raised questions about the ability of LES to realistically reproduce some unique features in the atmosphere. While there are individual LES studies of TCs, the sensitivity of LES to SGS parameterization has never been examined when the LES approach is used to simulate TCs. Such sensitivity tests are needed since intense turbulence in the eyewall can exist well beyond the PBL. In this study, we have tested three 3D SGS models commonly used in LESs: (a) 3D Smagorinsky SGS model (Smagorinsky, 1963), (b) 3D 1.5-order TKE SGS model (Deardorff, 1980), and (c) 3D NBA SGS model (Kosović, 1997).

It remains a mystery as to what the real value of vertical eddy exchange coefficients in the eyewall should be because of the difficulties to obtain vertical turbulent fluxes in the eyewall observationally. There are also difficulties to calculate vertical turbulent fluxes from the LES output. One of them is how to appropriately define the mean of a variable. For fast responding in-situ observations, the mean is commonly calculated as the average over a time period, and then, using the eddy correlation method to calculate the covariance of two variables. For classic LES applications in non-TC conditions, the domain-mean is often used when calculating vertical turbulent fluxes, which is appropriate as the ambient condition of the PBL is assumed to be horizontally homogeneous. However, such a method cannot be extended to LES of a TC as the fields of a storm vortex are not horizontally homogeneous. If a mean would include both violent eyewall and peaceful eye, the estimated covariance would be exaggerated. Furthermore, if the eddy correlation method is applied to the entire LES domain, then, one would only obtain one vertical profile of eddy exchange coefficient. It would be incorrect to apply this vertical profile to both eyewall and eye as the turbulent mixing in these two regions is completely different. One way to solve this problem is to define a sub-domain centered at each model grid, and then, use the LES output in the sub-domain for vertical flux calculation at each grid using eddy correlation method via,

$$F_\varphi = \overline{w'\varphi'} = \overline{(w - \overline{w})(\varphi - \overline{\varphi})} \tag{7}$$

where $\varphi$ is a generic scalar, $F_\varphi$ is the vertical flux of $\varphi$ at each grid, $w$ is the vertical velocity, overbar and prime indicate the average over the sub-domain and the deviation away from the average, respectively. In the first-order closure, the vertical momentum flux components may be represented as,

$$\overline{w'u'} = -k_m \frac{\partial \overline{u}}{\partial z}, \qquad \overline{w'v'} = -k_m \frac{\partial \overline{v}}{\partial z}, \tag{8}$$

where $k_m$ is the eddy exchange coefficient of momentum, $\frac{\partial \overline{u}}{\partial z}$ and $\frac{\partial \overline{v}}{\partial z}$ are the vertical gradient of mean wind components over the sub-domain. In the eyewall, the non-local mixing induced by the convective eddies (or cells) generates a large amount of

up-gradient vertical fluxes, thus, to account for the up-gradient vertical transport in the first-order closure, the momentum eddy exchange coefficient is calculated as,

$$K_m = \tau / \sqrt{(\frac{\partial \bar{u}}{\partial z})^2 + (\frac{\partial \bar{v}}{\partial z})^2} \qquad (9)$$

where $\tau = (\overline{w'u'}^{\,2} + \overline{w'v'}^{2})^{\frac{1}{2}}$ is the total vertical momentum fluxes.

Another important thing that needs to be considered is how large the sub-domain should be because the size of a sub-domain determines the contributions to the vertical fluxes from different scales of resolved eddies by LES. The horizontal grid resolution of HWRF-v3.8a is 2 km, meaning that eddies with scales greater than 2 km are resolved by HWRF. What need to be parameterized by HWRF PBL scheme are the vertical transport induced by eddies smaller than 2 km. Thus, in this study a 2 x 2 km$^2$ box is used as the sub-domain for vertical flux calculation at each grid point. Figure 4a shows the azimuthal-mean

radius-height distribution of the total vertical momentum fluxes, $\tau$, induced by the resolved eddies with scales smaller than 2 km from the LES run that uses the 3D NBA SGS model. The vertical profiles of eddy exchange coefficients of momentum from the three LESs that use different SGS models averaged over the radii of 30 - 60 km (where the eyewall is located) are shown in Fig. 4b. Note that the results shown in the figure have been averaged over 3 – 8 simulation hours and the SGS eddy exchange coefficients are the direct output from the SGS models. It clearly shows that the strong vertical momentum fluxes

induced by the resolved eddies keep increasing with height and reach the peak above the PBL (defined in the conventional way) in the low troposphere, and then, extend all the way up to the upper troposphere in the eyewall. There is no discontinuity across the PBL that separates the turbulent transport generated by the surface turbulent processes and cloud turbulent processes aloft in the eyewall. The resolved eddy exchange coefficients in the eyewall appear to be large and dominate the SGS coefficients. This is mainly caused by the limitation of using down-gradient parameterization of the first-

order closure to represent non-local mixing in the eyewall where the combined effects of the large up-gradient vertical transport and small vertical gradient of mean variables lead to the large eddy exchange coefficient.

    The discussion above and the results shown in Figs. 2, 3 and 4 suggest that to appropriately parameterize the turbulent mixing in the eyewall and rainbands, one may have to abandon the idea of using the diagnosed "PBL" height to artificially separate the internally connected turbulence generated by the PBL and cloud processes. From the nature of turbulent mixing,

it is more logical to treat the turbulence in the eyewall and rainbands generated by the different processes as a whole, i.e., treat the entire turbulent layer (TL) as an integrated layer. Physically, it makes sense as turbulent mixing generated by different processes in a deep convective environment cannot be artificially separated. It is important to point out that such a change from "PBL" to "TL" will not affect the turbulent mixing parameterization outside the deep convective area since the "TL" is virtually the same as the "PBL" in that case. The remaining question is how to appropriately define and determine a

"TL" in the eyewall and rainbands.

One way to improve the representation of turbulent mixing in the eyewall and rainbands is to develop a physically robust scheme using more sophisticated approaches, such as, TKE, high-order, or nonlocal closure, to replace Eqs. (5) and (6) to calculate vertical eddy exchange coefficient. However, a sophisticated method may not necessarily generate the desired results without significant tuning effort and thorough evaluation against observations, since an operational model consists of many physics modules that interact with each other and with the model dynamic core. How to integrate an individual scheme in a model to work in concert with other modules is an important but difficult scientific and technical problem. Moreover, the low vertical resolution above the PBL due to the stretching vertical grids commonly used in models makes it even more difficult to parameterize in-cloud turbulence above the PBL. To avoid possible degrading of HWRF's forecasting performance, a practical way is to keep the current framework of PBL scheme and refine it by incorporating an in-cloud turbulent mixing parameterization with the existing PBL scheme in a unified matter. Technically, this is relatively easy to do and scientifically it makes sense, since the "TL" should be the same as the "PBL" outside deep convective regions, and thus, nothing needs to be changed for the current PBL scheme used in HWRF. The only change that needs to be made is to overwrite the default diagnosed PBL height in the eyewall and rainbands with a newly determined "TL" height.

Since this study focuses on the turbulence generated by the cloud processes, a simple way to determine "TL" is to link "TL" directly to model-predicted cloud properties. A natural choice of such cloud properties is the cloud radar reflectivity, a product normally available from the microphysics module of a model. In the operational HWRF version 3.8a, the Ferrier-Aligo microphysical scheme (Aligo et al. 2018) calculates radar reflectivity at each time step. Figure 5 shows an example of the horizontal spatial distribution of HWRF simulated cloud radar reflectivity at different altitudes for Hurricane Jimena (2015) at an instant time along with an individual vertical profile of radar reflectivity in the eyewall and azimuthal-mean radius-height distribution of radar reflectivity. The vertical profile clearly shows that the simulated radar reflectivity in the eyewall remains nearly constant with height below the freezing level, and then, decreases sharply around 6 – 7 km in altitude. This unique feature allows us to determine "TL" from the radar reflectivity under the assumption that "TL" is virtually the cloud layer with prevalence of turbulence. After many tests, we choose 28 dBZ as a critical value to define "TL" in HWRF simulations. If no such a layer with radar reflectivity consistently greater than 28 dBZ is found or such defined "TL" is lower than the default "PBL", then, the default "PBL" is assumed to be the "TL". Thus, the change from "PBL" to "TL" will not affect the treatment of turbulent mixing elsewhere except for the diagnosed eyewall and rainbands with large reflectivity. Once "TL" is determined, the eddy exchange coefficients below and above the top of the diagnosed "TL" will be calculated following Eqs. (5) and (6), respectively. To retain the HWRF predicted turbulent structure and transport within the PBL, the eddy exchange coefficients below the PBL height are, then, overwritten by the eddy exchange coefficients determined by the default diagnosed "PBL" with a smoothing applied at the top of the "PBL" so that the eddy exchange coefficients in the eyewall and rainband change continuously from the PBL to the cloud layer. Thus, nothing is changed for the HWRF PBL scheme except that the new scheme includes an in-cloud turbulent mixing parameterization in the eyewall and rainbands determined from the "TL".

Note that such defined "TL" does not include the turbulence generated in the anvil clouds in the upper troposphere where the eyewall upward flow turns outward, becoming outflow. Outside a convection regime, the anvil clouds are detached from the PBL in model vertical columns, thus, "TL" concept does not apply. According to Emanuel (2012)'s "self-stratification" intensification hypothesis, the turbulence in the outflow is important because it acts to set the thermal stratification of the

outflow. The resultant gradients of outflow temperature provide a control of an intensifying vortex. In their analyses (Emanuel and Rotunno 2011; Emanuel 2012), the instability for generating small-scale mixing in the outflow was estimated by the gradient Richardson number. However, since numerical models use stretching grids in the vertical, it is very difficult to parameterize the SGS turbulent mixing in the outflow regions using bulk Richardson number at a very low vertical resolution. Moreover, since the main focus of this study is on the turbulent mixing above the PBL generated by cloud

processes within the convective eyewall and rainbands, we want to isolate this problem from the complication of the outflow turbulence. For these reasons, the effect of outflow turbulence on the intensification process will not be discussed in this study.

Figure 6 shows the horizontal distribution of the simulated eddy exchange coefficients of momentum, $k_m$, at different altitudes of Hurricane Jimena (2015) by the HWRF with the inclusion of an in-cloud turbulent mixing parameterization

along with the azimuthal-mean radius-height distribution of $k_m$. Compared with Fig.2, the modification from "PBL" to "TL" allows HWRF to successfully capture the in-cloud turbulent mixing. The horizontal spatial distribution of $k_m$ above the PBL well reflects the eyewall and rainband structure of the TC vortex, which is in stark contrast to the default operational HWRF that generates virtually no turbulent mixing above the PBL (Fig. 2). However, the peak of the parameterized $k_m$ appears to be smaller than that from the LESs (Fig. 4b). Note that this difference may result partially from the uncertainty in determination

of vertical fluxes using LES output as we pointed out previously and partially from the crude method to treat in-cloud turbulence. As we stated previously, our method itself does not consider the specific mechanisms in generating in-cloud turbulence, and thus, the scheme in its current form may not be directly used in operational forecasts. Nonetheless, this simple modification allows us to look into and examine the role of eyewall and rainband SGS eddy forcing above the PBL in TC intensification. One advantage of the change from "PBL" to "TL" is to allow for a possible internal interaction between

microphysics and turbulence. In real TCs, cloud microphysical processes directly interact with in-cloud turbulence to generate the diabatic heating that drives the overturning circulation. The negligible turbulent mixing above the PBL in the operational HWRF virtually removes the microphysics-turbulence interaction in eyewall/rainband clouds. While simple, the inclusion of an in-cloud turbulent mixing parameterization by overwriting the "PBL" height with the "TL" provides an avenue that allows microphysics to directly interact with turbulence in simulations. In the next section, we show that such a

modification improves HWRF's skills in predicting TC intensity change, in particular, RI.

## 3. Results

To evaluate the modified HWRF PBL scheme with the inclusion of an in-cloud turbulent mixing parameterization and investigate the role of eyewall/rainband eddy forcing in modulating TC intensity change, we simulated 16 storms in the Atlantic basin and eastern tropical Pacific in the past four seasons (2014-2017) with different intensities ranging from tropical storms to major hurricanes. For each storm, we simulated 4 cycles with the model initialized at different time. These simulations allow us to provide an initial evaluation of the in-cloud turbulent mixing parameterization and address scientific issues associated with TC intensity change in different TC conditions. In this paper, we mainly focus on RI. Here, we present one of the four simulations of Hurricane Jimena (2015), which was initialized at 12:00 UTC 27 August, 2015. Using this simulation, we investigate how eyewall and rainband eddy forcing modulates the RI of Jimena (2015).

Figure 7 compares the storm track and intensity from the two simulations of Jimena (2015) by HWRF using the default PBL scheme and the PBL scheme that includes an in-cloud turbulent mixing parameterization along with the best track data. These two simulations are named as "DEF-HWRF" and "TL-HWRF" respectively hereafter. While DEF-HWRF does an excellent job in reproducing the observed track, it under-predicts the observed storm intensity by a large margin. The integrated turbulent mixing parameterization in the eyewall and rainbands ("TL-HWRF") shows little impact on the simulated storm track but improves the intensity forecast substantially. It allows HWRF to successfully capture the observed RI of Jimena, suggesting the importance of eyewall/rainband turbulent mixing above the PBL in modulating TC intensification. To see if the resultant improvement in intensity simulation by "TL-HWRF" is mainly caused by the SGS eddy momentum transport or by eddy heat/moisture transport, two additional experiments were executed. In the first experiment, we only modified the eddy exchange coefficient for momentum $k_m$ while keeping the eddy exchange coefficient for heat and moisture $k_{t,q}$ the same as the default. We reversed such a change in the second experiment. As shown in Fig.7, both the modified turbulence closures for momentum alone and for heat/moisture alone show non-negligible impacts on TC intensification. This result is not unexpected. While the tangential eddy forcing for momentum directly involves in the acceleration or deceleration of the primary circulation of a TC, the thermodynamic eddy forcing is sufficiently strong to modulate the secondary overturning circulation that interacts with the primary circulation during TC evolution. Note that in HWRF the eddy exchange coefficients for heat and moisture are treated as the same, thus, we did not further separate them in our study. In the following sections, we explore and discuss the underlying reasons for such an improvement in intensity forecast.

Figure 8 shows the Naval Research laboratory 37 GHz color image from the Advanced Microwave Scanning Radiometer 2 (AMSR2) at 20 UTC 28 August 2015, a time close to the initiation of Jimena's RI. A well-defined inner-core structure including a quasi-closed ring feature around the storm center (somewhat broken in the northwest quadrant) is clearly visible in the satellite image. From a large amount of 37 GHz microwave color products, Kieper and Jiang (2012) showed that the

first appearance of a cyan color ring around the storm center is highly correlated to subsequent RI, provided that environmental conditions are favorable. This result is consistent with the later analyses of Tropical Rainfall Measuring Mission (TRMM) 29 Precipitation Radar (PR) data (Jiang and Ramirez 2013; and Tao and Jiang 2015), which showed that nearly 90% of RI storms in different ocean basins formed a precipitation ring around the storm center prior to RI. The relationship between the ring feature and the subsequent RI obtained from these observational studies is consistent with the theoretical finding of Nolan et al. (2007), who demonstrated that the intensification processes of a balanced, baroclinic TC-like vortex is mainly driven by the TC symmetric response to the azimuthally-averaged diabatic heating, rather than to the heating directly associated with individual asymmetries distributed around the TC vortex. To see if Jimena's RI possesses the similar RI signature found in these observational and theoretical studies, we carefully examined the inner-core structure of the simulated Jimena (2015) prior and during the early stage of RI. Figure 9 shows the horizontal distribution of simulated vertical velocity and hydrometeor mixing ratio at 5 km altitude from the two HWRF simulations with and without an in-cloud turbulent mixing parameterization. The vortex inner-core structure in "DEF-HWRF" is poorly organized and the simulated eyewall appears to be much larger in size than the satellite observed eyewall (Figs. 9a and 9c). It suggests that HWRF with operational model physics is unable to generate the right vortex inner-core structure needed for the subsequent RI. In contrast, "TL-HWRF" produces a well-defined quasi-closed ring around the storm center that is clearly shown in both dynamic (Fig. 9b) and thermodynamic (Fig. 9d) fields. The size of the simulated quasi-closed ring in "TL-HWRF" is similar to that shown in the satellite image. In addition, the simulated asymmetric rainband structure with the strongest convection occurring in the southeast quadrant is consistent with the satellite observation. The similar vortex inner-core structure shown in both satellite observations and "TL-HWRF" simulation implies that the RI of Jimena (2015) is likely governed by the axisymmetric dynamics similar to what was found by Vigh et al. (2018) who showed that some of the VRI and ERI storms, such as Hurricane Patricia (2015), Typhoon Meranti (2016), and Hurricane Maria (2017), can be well captured by the axisymmetric CHIPS with zero-wind shear. The fact that the observed TC inner-core structure including the quasi-closed ring feature is reproduced by "TL-HWRF" but not by the default HWRF suggests that the SGS physics involving with the in-cloud turbulent mixing above the PBL facilitates the realization of the axisymmetric dynamics underlying the RI of TCs in 3D full-physics simulations.

Figure 10 shows the simulated azimuthal-mean radius-height structure of vertical velocity, hydrometeor mixing ratio, radial inflow/outflow, and radial flow convergence averaged over the RI period from 06 UTC 28 to 06 UTC 29 August, 2015. Compared with the "DEF-HWRF", "TL-HWRF" generated much stronger updrafts (thick gray contours) in the eyewall, stronger radial inflow (red contours) within the PBL, and outflow (white contours) above, which are consistent with the strong storm intensity simulated by this experiment (Fig. 7). Furthermore, in the "TL-HWRF" experiment, the radial flow convergence (black contours) matches well with the eyewall updrafts. This feature facilitates an efficient transport of moisture into the eyewall to result in a large amount of condensate (color shading) in the eyewall. The resultant latent heating fosters the rapid converging spin-up processes as air parcels move radially inward and ascend swiftly within the

eyewall. This result suggests the importance of microphysics-turbulence interaction in TC intensification. In contrast, the peaks of persistent radial flow convergence in "DEF-HWRF" do not occur in the eyewall, but rather extend radially outward along the interface of radial inflow and outflow. Such a structure is apparently unfavourable to the rapid development of the vortex, since it cannot generate the efficient converging spin-up processes. Therefore, the simulated storm intensity difference by the two HWRFs may be largely attributed to the differences in the strength and structure of the secondary overturning circulation in this case. However, we note the depth of the radial inflow layer is similar in both HWRF simulations. It suggests that the inclusion of an in-cloud turbulent mixing parameterization aloft in the simulation does not alter the basic structure of the PBL in the TC vortex inner-core region.

To better understand the intensification processes in the two HWRF simulations, we examined the tangential eddy forcing ($F_\lambda + F_{sgs\_\lambda}$) for the primary circulation of the TC vortex. The model-resolved tangential eddy forcing $F_\lambda$ is calculated by Eq. (1) using the wind fields in the standard HWRF output. As we noted previously, in this study we only focused on the vertical turbulent mixing, therefore, the SGS tangential eddy forcing $F_{sgs\_\lambda}$ diagnosed here is only the one calculated from the tendencies directly generated by the vertical turbulent mixing scheme (or PBL scheme). The SGS eddy forcing resulting from horizontal diffusion is not included. Figure 11 compares the SGS tangential eddy forcing averaged over the RI period from 06 UTC 28 to 06 UTC 29 August, 2015 between the two HWRF simulations, where the upper and bottom panels show the azimuthal-mean radial-height structure of SGS tangential eddy forcing and its horizontal plane view at 3 km altitude, respectively. There are a couple of interesting features shown in the figure. First, the radial-height structure of SGS tangential eddy forcing generated by "DEF-HWRF" (Fig. 11a) is very similar to that from a 3D full-physics TC simulation shown in Persing et al. (2013, their Figs. 10f &11f). The SGS eddy forcing above 2 km in the eyewall region is virtually zero because in-cloud turbulent mixing is not included in these simulations. In contrast, the in-cloud turbulent mixing parameterization in "TL-HWRF" allows HWRF to successfully generate the SGS tangential eddy forcing associated with the eyewall and rainband convection above the PBL (Fig. 11b). Such a SGS eddy forcing in the eyewall region from the layer just above the PBL to the upper troposphere has not been shown and discussed in previous numerical studies. Second, in addition to the expected strong negative SGS tangential eddy forcing within the PBL, the in-cloud turbulent mixing parameterization generates an interesting vertical structure of SGS tangential eddy forcing above the PBL in the eyewall region. Although it is much weaker than that in the PBL, the SGS tangential eddy forcing in the eyewall does show positive values at the heights just above the inflow layer as well as above the mid troposphere, suggesting that the eyewall SGS tangential eddy forcing above the PBL is indeed involved in the vortex spin-up processes during the RI. What remains unclear is the fidelity of the parameterized SGS eddy forcing above the PBL and its sensitivity to specific turbulent mixing parameterization. This constitutes one of the uncertainties in storm intensity simulation.

The model-resolved tangential eddy forcing averaged over the RI period from 06 UTC 28 to 06 UTC 29 August, 2015 is shown in Fig. 12. The basic radial-height structures of the resolved eddy forcing generated by the two simulations are similar

to a certain extent, and share similar features to those from Persing et al. (2013)'s 3D full-physics TC simulation (cf. their Figs. 10g &11g). But the resolved eddy forcing in "TL-HWRF" is much stronger than that in "DEF-HWRF". A robust feature shown in both simulations is the positive eddy forcing right above the inflow layer in the vicinity of the eyewall. From the perspective of absolute angular momentum conservation, this positive tangential eddy forcing is directly linked to the vortex spin-up. But currently we have little knowledge on what determines the sign, magnitude, and vertical structure of eddy forcing. Future research should focus on elucidating these issues regarding how eyewall and rainband eddy processes regulate the TC intensification.

Comparing Fig. 12b with Fig. 11b, it is easy to see that the model-resolved eyewall eddy forcing above the PBL in the "TL-HWRF" experiment has a magnitude about 5 times larger than the corresponding SGS eddy forcing, suggesting that the resolved eddy processes provide a major forcing that drives the primary circulation of the TC vortex in this case. As model resolution keeps increasing, we expect that the resolved eddy forcing will become more dominant. This is certainly a promising result, implying that numerical forecast of TC intensification may be ultimately a resolution problem. The difficulty, however, stems from the strong dependence of model-resolved eddy forcing and TC inner-core structure on the parameterized SGS eddy processes at the current resolution. As we showed in Figs. 9, 10, and 12, the only modification in SGS turbulent mixing parameterization above the PBL in the eyewall and rainbands result in substantial differences in the vortex structure, secondary overturning circulation, and model-resolved eyewall/rainband eddy forcing. Such a dependence of model-resolved TC fields on the parameterization of SGS in-cloud turbulence above the PBL is currently not well understood. It could stem from the fact that the large energy-containing turbulent eddies, such as kilometre and sub-kilometer convective elements or roll vortices (evidenced in the LESs), are not resolved by the current model resolution of 2 km, and could also result from the dynamical-microphysical interaction in TC clouds. The strong dependence of the resolved TC vortex on SGS parameterization poses a great challenge for accurate prediction of TC intensity change.

The results presented previously show that eyewall/rainband eddy forcing plays a key role in Jimena's RI and the inclusion of parameterization of eyewall/rainband in-cloud turbulent mixing above the PBL substantially improves HWRF's skills on generating robust eddy forcing for accurate intensity prediction. Such an improvement is not a special case, but is shown in HWRF simulations of other major TCs as well. Figure 13 shows the HWRF simulated maximum wind speed and storm central pressure of four other major hurricanes compared with the best track data. In all cases, the intensity simulations were improved due to the inclusion of an in-cloud turbulent mixing parameterization, in particular, "TL-HWRF" was able to partially capture the observed RI of Harvey (2017) and Marie (2014), which was largely missed by "DEF-HWRF". Similar to the HWRF simulations of Jimena (2015), our analyses show that the better intensity forecasts of these storms by "TL-HWRF" can be largely attributed to the improved simulation of storm inner-core structure and eyewall/rainband eddy forcing needed for TC vortex spin-up. As another example, Figure 14 compares the satellite observed vortex inner-core structure of Harvey (2017) with the simulated ones by the two HWRFs during the early and middle stages of Harvey's RI.

The asymmetric rainband structure, the size and the structure of the eyewall shown in satellite observations are reasonably reproduced by "TL-HWRF". But "DEF-HWRF" was not able to simulate the observed inner-core structure; in particular, the simulated eyewall is poorly defined and the size is much larger than the observed one. This result once again suggests that at the current model resolution the realization of axisymmetric dynamics underlying RI of TCs is sensitive to the parameterization of in-cloud SGS eddy processes above the PBL in the eyewall and rainbands in 3D full-physics simulations.

Our testing simulations also show that the inclusion of an in-cloud turbulent mixing parameterization in the eyewall and rainbands does not appear to degrade HWRF's performance on those cases that operational HWRF has decent forecasts on or generate false RI for those weak storms. As an example, Figure 15 shows the storm intensity of Hermine (2016) simulated by the two HWRFs compared with the best track data. Hermine (2016) is a weak storm with the peak intensity just reaching Category-1 hurricane strength. The simulation results show that the integrated turbulent mixing parameterization in the eyewall and rainbands only has a marginal impact on the HWRF predicted storm intensity. It did not over-predict storm intensity or generate false RI that one may be concerned about. We have worked with the Environmental Modeling Center (EMC), NOAA, to implement our modified PBL scheme in 2018 operational HWRF and tested it in operational HWRF full cycle simulations. The preliminary results from total 1,079 case simulations for various forecast lead times show that the modified HWRF noticeably reduces the bias error of maximum wind speed (Fig. 16). Currently, we continue working with EMC to improve and refine the parameterization of in-cloud turbulent mixing in the eyewall and rainbands.

## 4. Summary

Asymmetric eddy processes provide an important forcing for the evolution of the primary and secondary circulations of a TC. Because of the discrete grids used in numerical models, the eddy forcing with a continuous spectrum is split into two parts resulting from the model-resolved and parameterized SGS eddy processes. While higher model resolution allows the model-resolved eddy forcing to be better resolved, the parametric determination of SGS eddy forcing is source of uncertainty in storm intensity prediction.

In numerical simulations, the SGS eddy forcing is determined by the turbulent mixing scheme. Turbulence is commonly regarded as a flow feature of the PBL. In fair-weather conditions the turbulent PBL is often cleanly separated from the free atmosphere above by a capping inversion. Except for occasional clear-sky turbulence, turbulent mixing is negligible above the PBL. The various PBL schemes used today in the state-of-the-art numerical models were designed to best represent the turbulent transport within the PBL. In a TC environment, however, turbulence is no longer solely generated by the shear production and buoyancy production associated with the PBL processes. Intense turbulent mixing can also be generated by cloud processes above the PBL in the eyewall and rainbands due to radiative cooling, evaporative cooling, and inhomogeneous diabatic heating and cooling. While the concept of PBL is still applicable in the eyewall and rainbands as to

the layer that is directly affected by the surface turbulent processes, the treatment of turbulent mixing must go beyond the conventional scope of the PBL. This is particularly true in the TC inner-core region as air parcels ascend swiftly within the eyewall and rainbands where there is no physical interface that separates the turbulence generated by the PBL processes and cloud processes aloft. The conventional PBL theory that treats the PBL as a shallow layer adjacent to Earth's surface becomes insufficient to explain the observed intensity change in some TCs. Such a deficiency of classic PBL theory is reflected in the PBL scheme used in HWRF. The HWRF PBL scheme is a typical first-order K-closure scheme that parameterizes turbulent mixing based on the diagnosed PBL height. Our analyses show that an artificial separation of the PBL from the free atmosphere above cannot appropriately represent the vertical turbulent structure and transport in the eyewall and rainbands, in particular, the simple method of parameterizing turbulent mixing above the PBL based on the bulk Richardson number is unable to account for the intense turbulent mixing aloft generated by eyewall/rainband cloud processes. As a result, the HWRF PBL scheme fails to generate the eyewall/rainband SGS eddy forcing associated with cloud processes above the PBL.

In this study, we developed a method to allow for an integrated turbulent mixing parameterization in the eyewall and rainbands based on the "TL" determined by the simulated radar reflectivity. Such a change from "PBL" to "TL" will not affect the turbulent mixing parameterization outside the eyewall and rainbands since the "TL" is virtually the same as the "PBL" in non-convective regions. This simple adjustment allows HWRF to successfully generate eyewall/rainband SGS eddy forcing above the PBL. Numerical tests on multiple major hurricanes show that the inclusion of an in-cloud turbulent mixing parameterization notably improves HWRF's skills on predicting TC intensity change, in particular, RI in several cases. While the performance of the modified turbulent mixing scheme is promising, our treatment of in-cloud turbulent mixing is very crude, and thus, the scheme may not be ready for use in operational TC forecasts in its current form. Nonetheless, our results show that numerical simulations of TC intensification are sensitive to the parameterization of SGS turbulent mixing induced by the cloud processes above the PBL in the eyewall and rainbands. Future research should focus on developing physically robust scheme to better represent in-cloud turbulent processes in 3D full-physics models and advance our theoretical understanding of how eyewall/rainband eddy forcing above the PBL modulates TC intensification including RI. There are scientific questions that need to be further addressed and clarified, such as, what determines the sign, magnitude, and vertical distribution of eyewall/rainband forcing? And is eddy forcing that leads to TC intensification a stochastic process or deterministic process?

While the improvement of TC intensity forecast due to the inclusion of an in-cloud turbulent mixing parameterization is clearly demonstrated, the underlying reason for such an improvement appears to be complicated. At first glance, the calculated SGS eddy forcing above the PBL is about five times smaller than the model-resolved eddy forcing (Figs. 11b and 12b). This would suggest that the model-resolved eddy forcing is the dominant forcing for the spin-up of the TC vortex at the current model resolution. However, the simulated TC inner-core structure, secondary overturning circulation, and the model-resolved eddy forcing show a strong dependence on the parameterized in-clouds SGS eddy processes above the PBL. The in-

cloud turbulent mixing parameterization appears to facilitate the realization of axisymmetric dynamical mechanism underlying RI of TCs in 3D full-physics simulations. These results suggest that the model-resolved and SGS eddy forcings are not independent, although they appear as two separate terms in the governing equations and are determined separately in numerical simulations. Such a dependence may result from the fact that the dynamical-microphysical interaction and large energy-containing turbulent eddies, such as kilometre and sub-kilometer convective elements and roll vortices, are not resolved but parameterized at a grid spacing of 2 km. Will further increasing of model resolution reduce the dependence of model-resolved fields on parameterized SGS processes? This question cannot be answered until the dynamical-microphysical interaction and large energy-containing eddies can be explicitly resolved. To do so, large-eddy resolution both horizontally and vertically is needed not only in the PBL (like classic LES) but also aloft in the eyewall and rainbands to resolve in-cloud turbulent eddies generated by cloud processes. This is not likely to happen in the near future for operational forecasts even with ever-increasing computational capability. Therefore, as model resolution keeps increasing, research effort should be continuously devoted to improving parametric representation of model physics not only in the PBL but also above the PBL to appropriately account for microphysical processes, in-cloud turbulent processes, and the interaction between microphysical and dynamical processes.

**Acknowledgement**

This work is supported by NOAA/HFIP under Grants NA14NWS4680030 and NA16NWS4680029, National Science Foundation under Grant AGS-1822238 and AGS-1822128, and BP/The Gulf of Mexico Research Initiative. We are very grateful to Dr. Michael T. Montgomery and an anonymous reviewer for their constructive and insightful comments, which lead to the improvement of the paper. Data used in this study can be accessed at http://vortex.ihrc.fiu.edu/download/HWRF-TUR/.

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

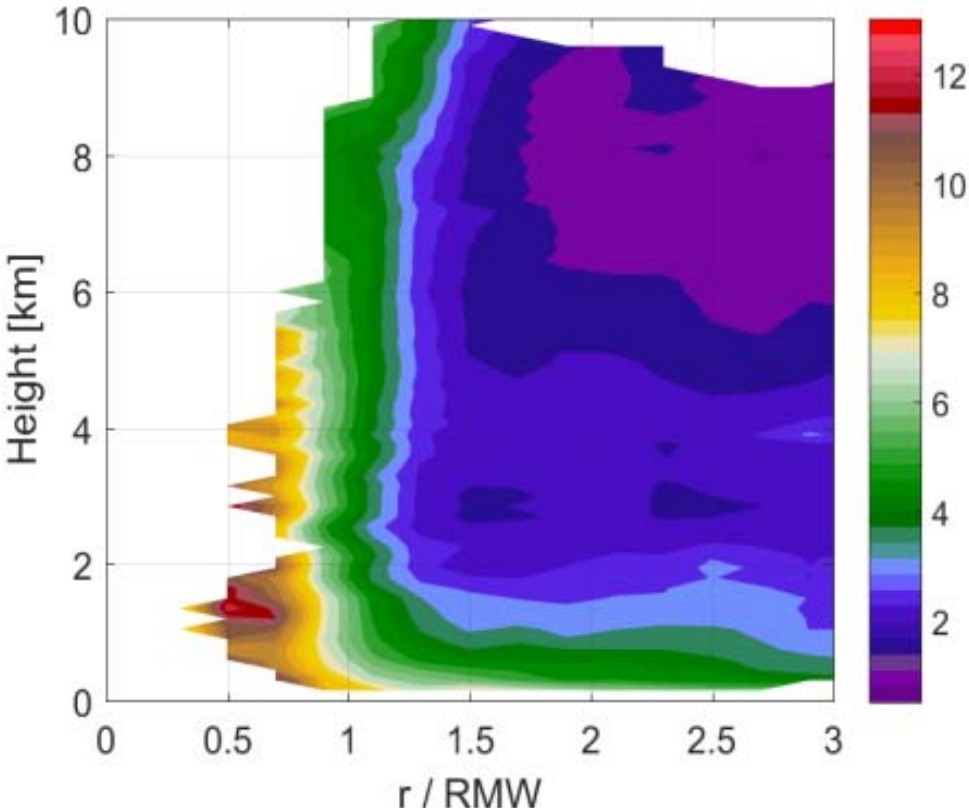

Figure 1: Composite TKE derived from airborne radar data from 116 radial legs of P3 flights in the 2003-2010 hurricane seasons as a function of height and the radius normalized by the radius of maximum wind (RMW).

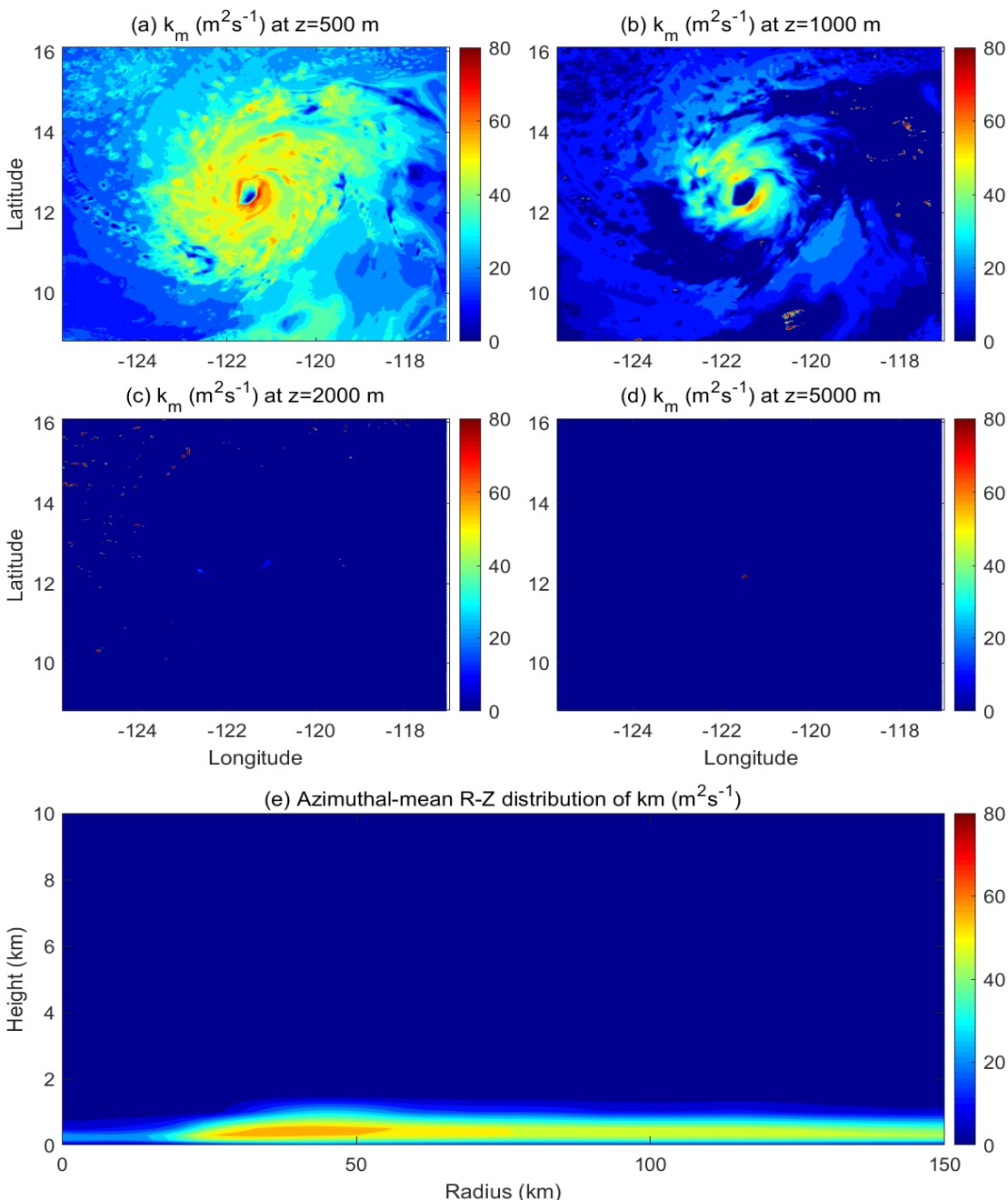

Figure 2: (a) – (d): Horizontal distribution of eddy exchange coefficients of momentum ($k_m$) at the altitudes of z = 0.5, 1.0, 2.0, and 5.0 km, respectively; (e): Azimuthal-mean radius-height distribution of $k_m$ from a HWRF simulation of Hurricane Jimena (2015) at 12:00 UTC 28 August, 2015.

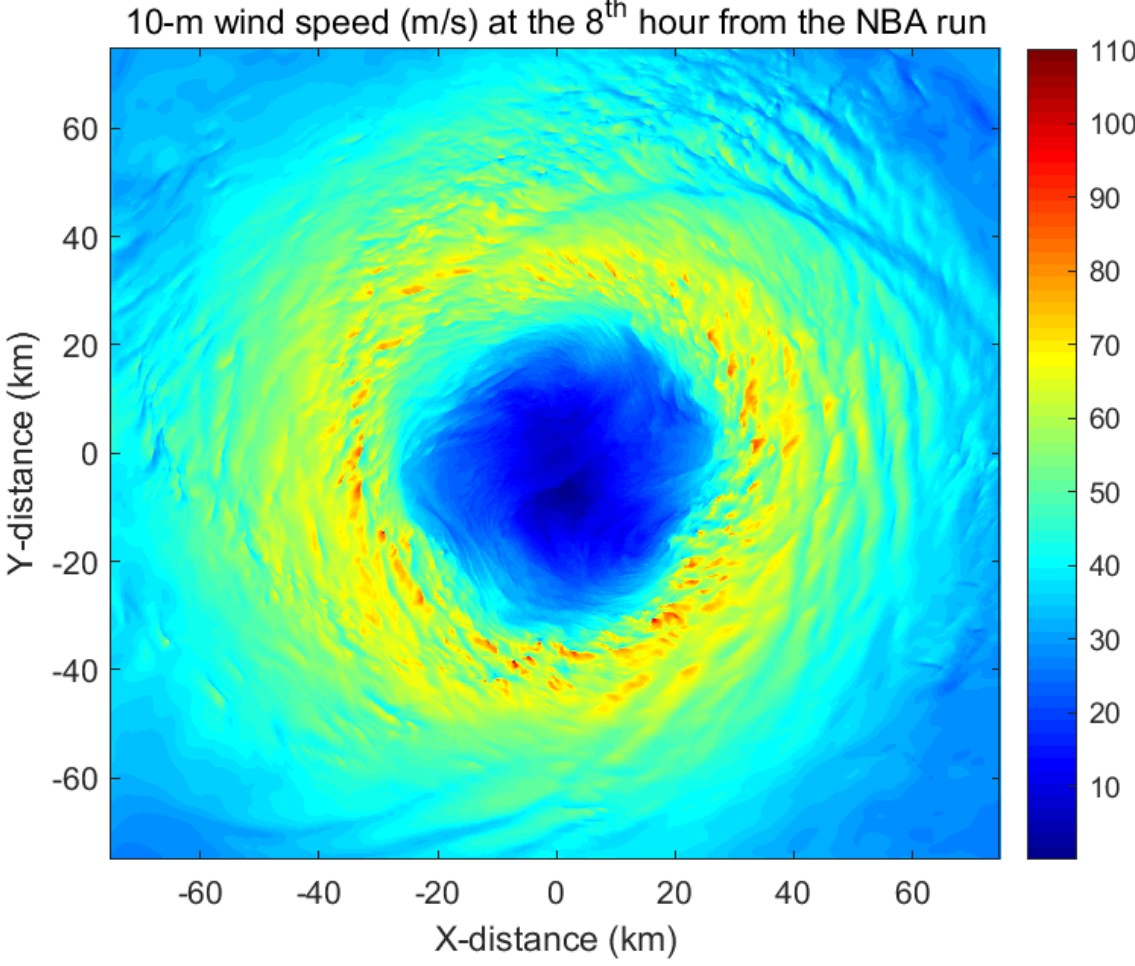

Figure 3: Instantaneous 10-m surface wind speeds of Hurricane Isabel (2003) at the 8th simulation hour by a WRF-LES that uses the 3D NBA SGS model.

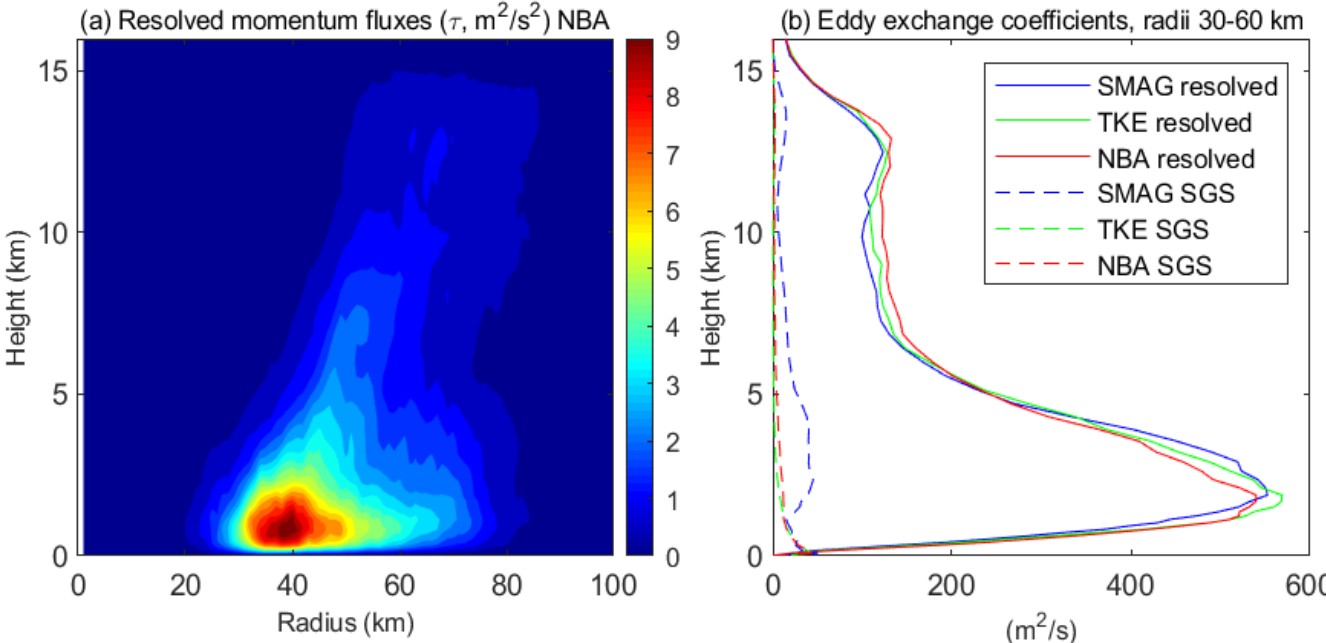

Figure 4: (a): Azimuthal-mean radius-height distribution of the vertical momentum fluxes, $\tau = (\overline{w'u'}^2 + \overline{w'v'}^2)^{\frac{1}{2}}$, induced by the resolved eddies with scales smaller than 2 km from the WRF-LES that uses the 3D NBA SGS model. (b): Vertical profiles of the parameterized (dashed) and resolved (solid) vertical eddy exchange coefficients of momentum averaged over 30 – 60 km radii (where the eyewall is located) from the three LESs that use different 3D SGS models. Note that the results are averaged over 3 – 8 simulation hours and the SGS eddy exchange coefficients are the direct output from the 3D SGS models used in the simulations.

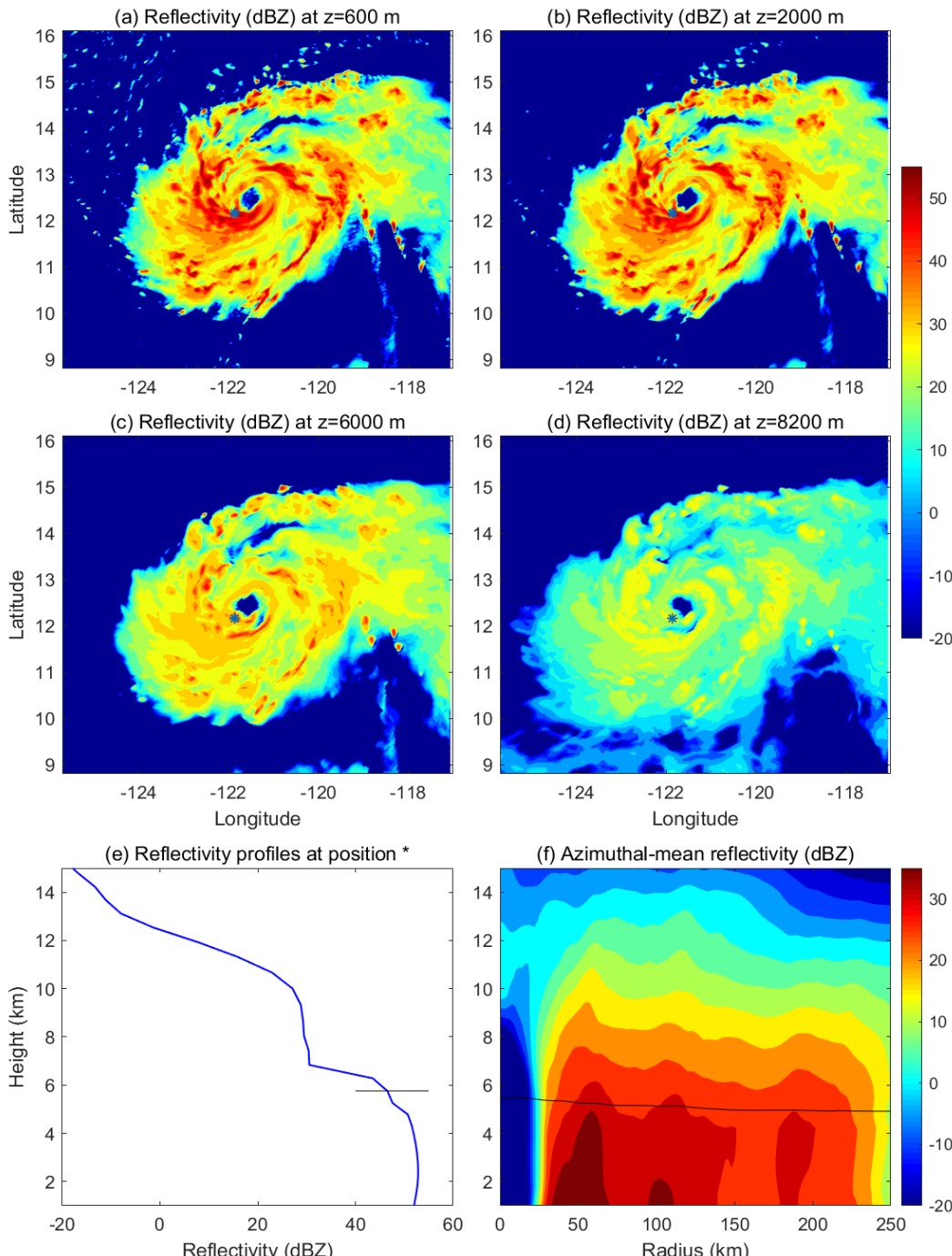

Figure 5: (a)-(d): HWRF simulated radar reflectivity of Hurricane Jimena (2015) at different altitudes (z = 0.6, 2.0, 6.0, and 8.2 km) at 12:00 UTC 28 August, 2015. (e): Vertical profile of radar reflectivity at a location in the eyewall marked by "*" in (a)-(d). (f): Azimuthal-mean radius-height structure of radar reflectivity. Black line in (e) and (f) indicates the freezing line.

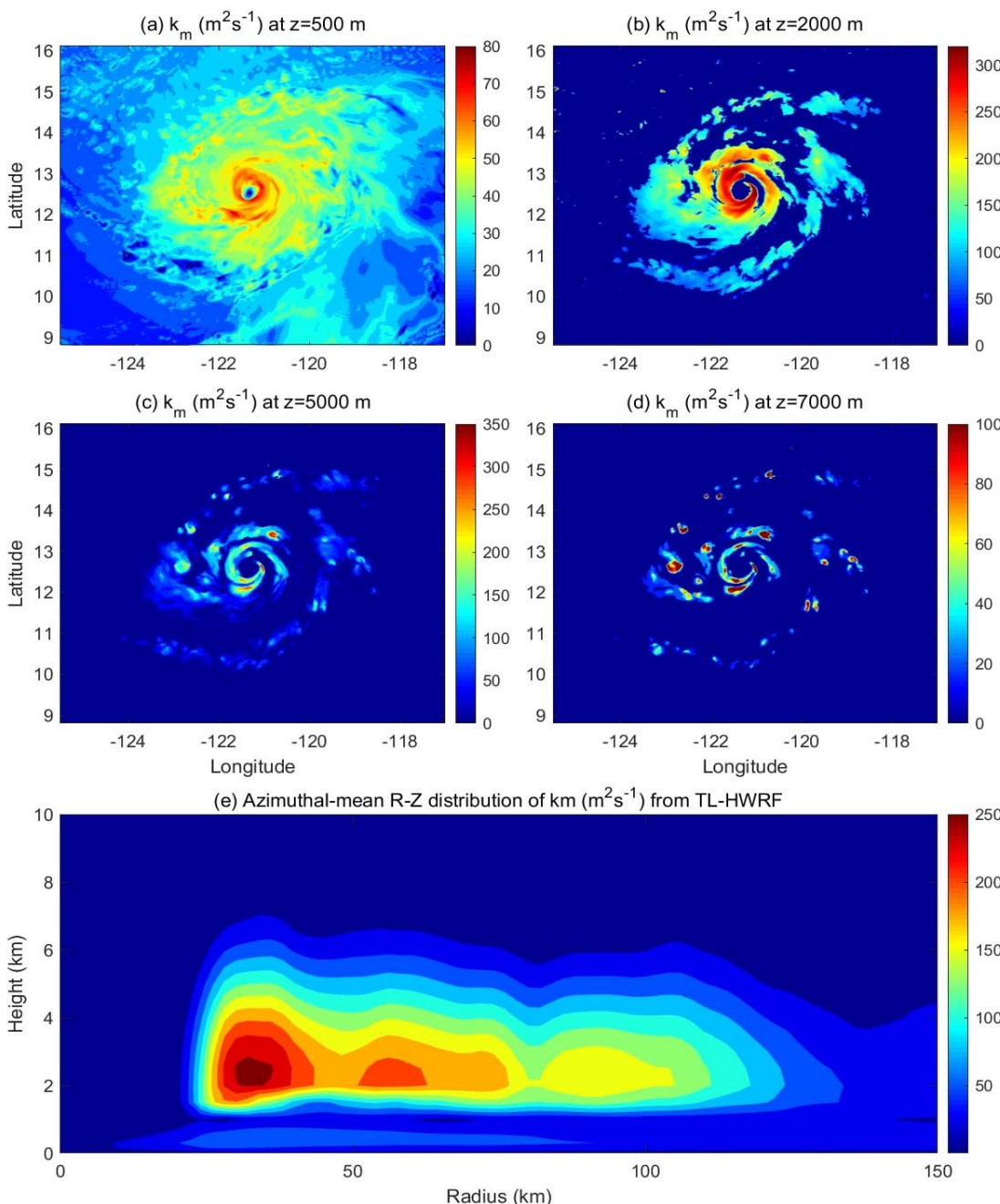

Figure 6: (a) – (d): Horizontal distribution of eddy exchange coefficients of momentum ($k_m$) at the altitudes of z = 0.5, 2.0, 5.0, and 7.0 km, respectively; (e): Azimuthal-mean radius-height distribution of $k_m$ from the HWRF simulation with the inclusion of an in-cloud turbulent mixing parameterization (TL-HWRF) at 12:00 UTC 28 August, 2015.

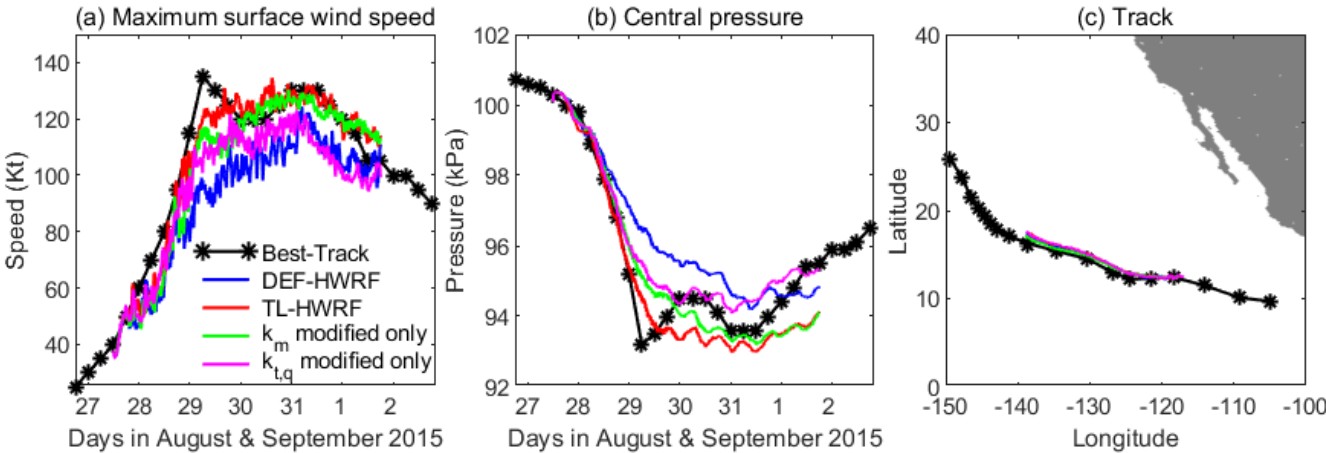

Figure 7: Comparison of HWRF simulated maximum surface wind speed, storm central pressure, and track of Jimena (2015) with the best track data (Black). Blue curve indicates the simulation by the default HWRF ("DEF-HWRF"). Red curve indicates the simulation by the HWRF with inclusion of an in-cloud turbulent mixing parameterization ("TL-HWRF"). Green curve represents the simulation in which only the eddy exchange coefficient for momentum is modified while keeping the eddy exchange coefficient for heat and moisture the same as the default. Magenta curve is opposite to the green curve in which only the eddy exchange coefficient for heat and moisture is modified.

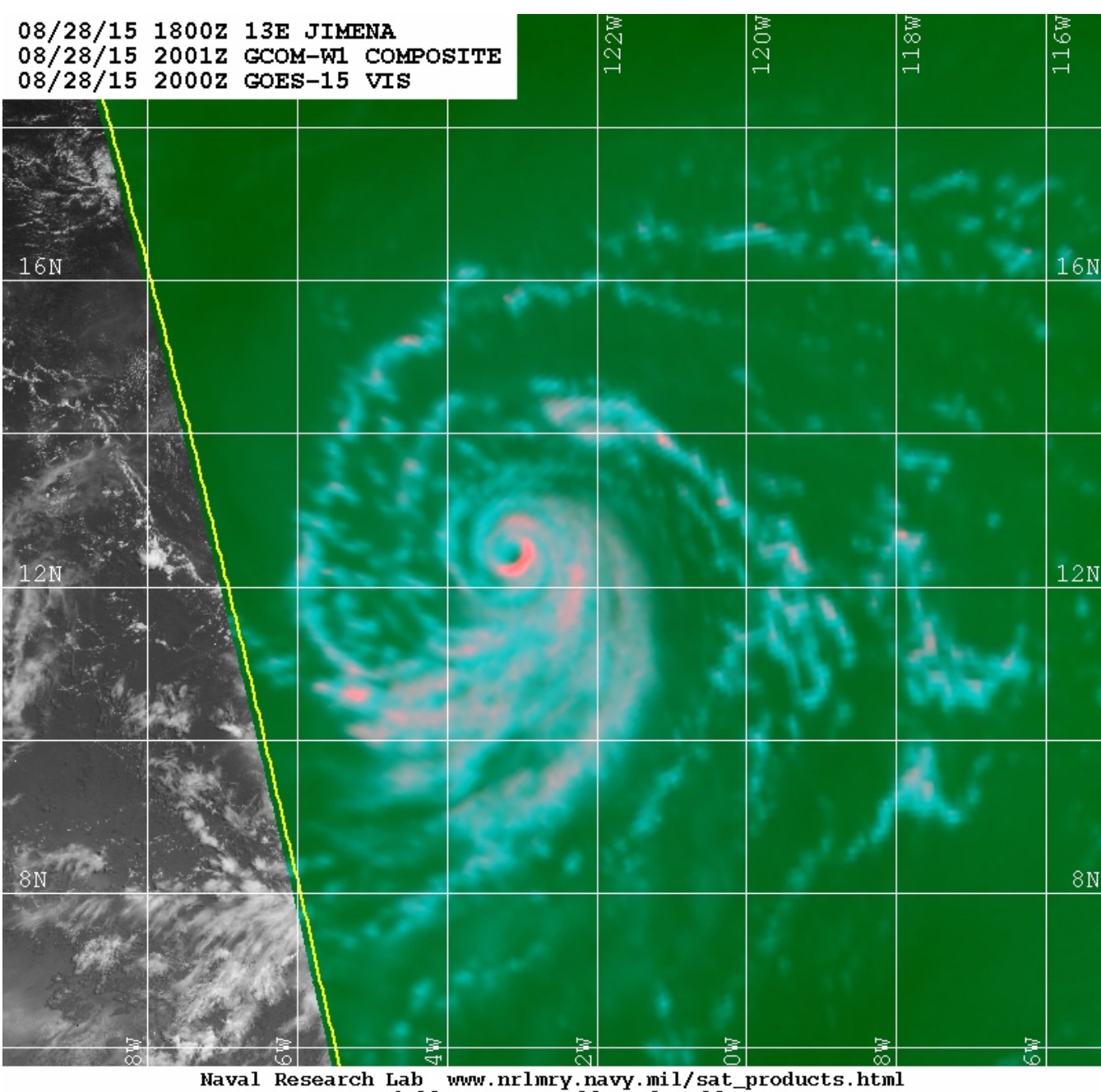

Figure 8: Naval Research Laboratory 37 GHz color image from the Advanced Microwave Scanning Radiometer 2 (AMSR2) at 20:00 UTC 28 August, 2015.

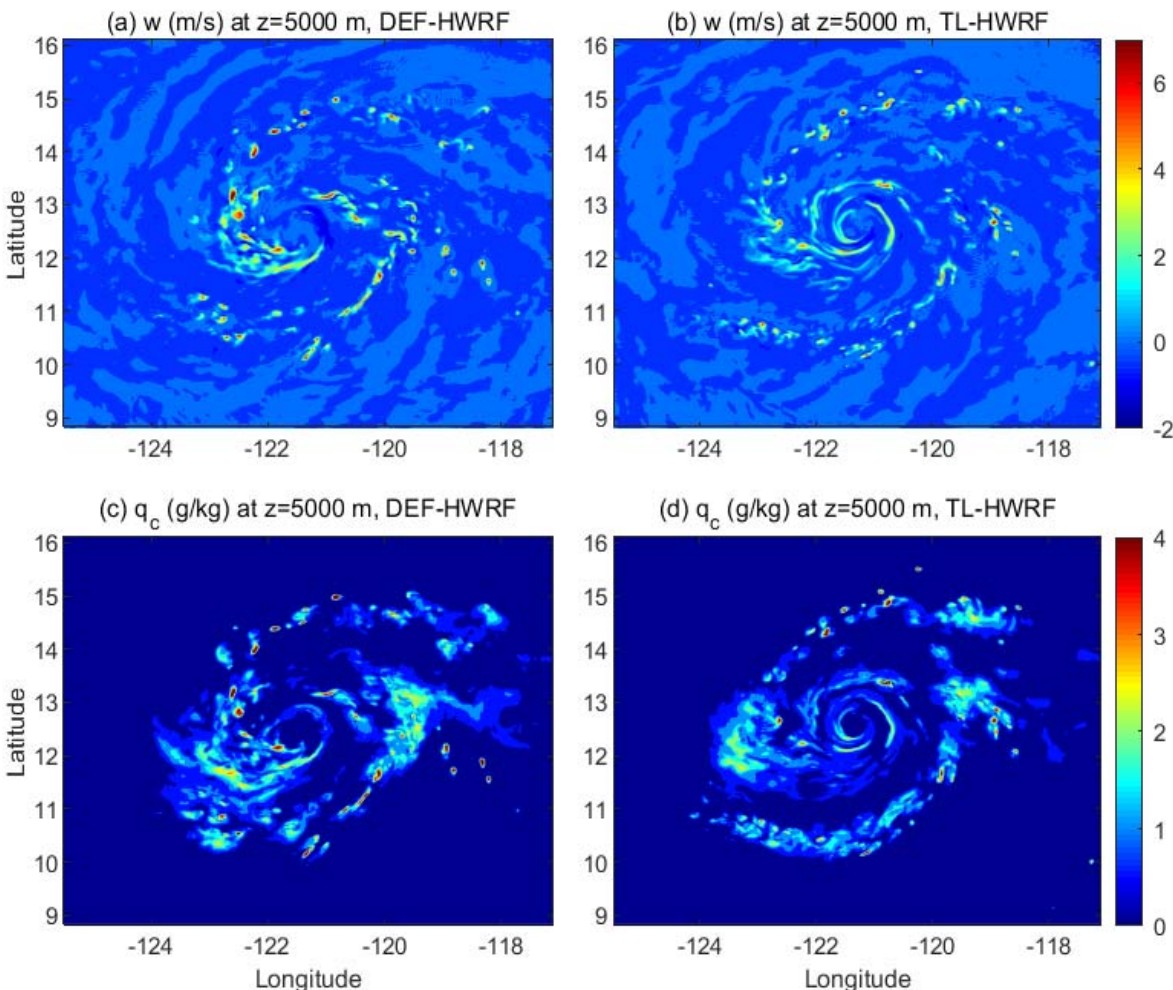

Figure 9: Simulated vertical velocity (ms⁻¹) and hydrometeor mixing ratio (gkg⁻¹) at 5.0 km altitude at 12:00 UTC 28 August, 2015 by the default HWRF (DEF-HWRF) and the HWRF with the inclusion of an in-cloud turbulent mixing parameterization (TL-HWRF).

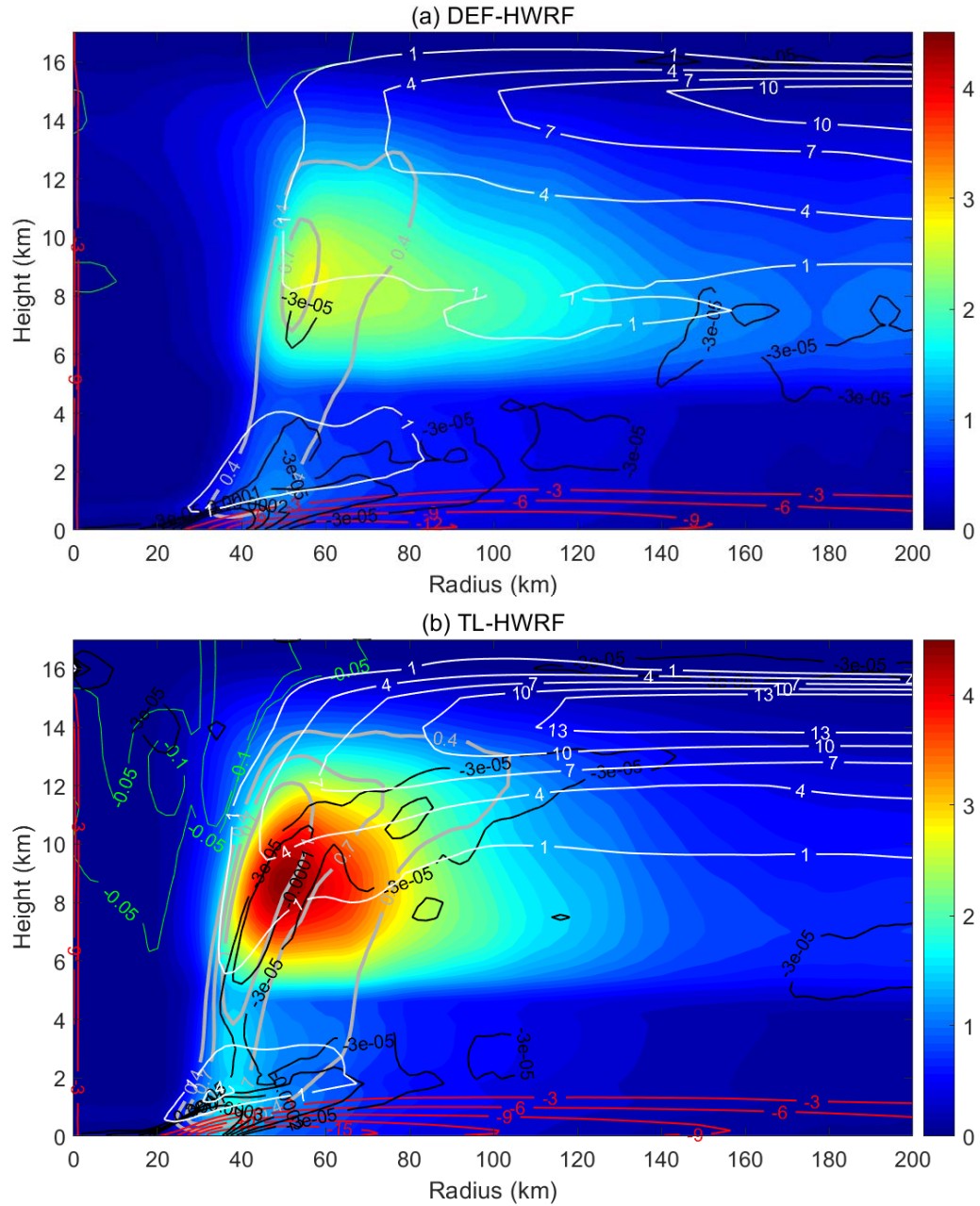

Figure 10: Simulated azimuthal-mean radius-height structure of updrafts (thick grey contours, ms$^{-1}$), downdrafts (green contours, ms$^{-1}$), hydrometeor mixing ratio (color shading, gkg$^{-1}$), radial inflow (red contours, ms$^{-1}$), outflow (white contours, ms$^{-1}$), and radial flow convergence (black contours, s$^{-1}$) averaged over Jimena's RI period from 06:00 UTC 28 to 06:00 UTC 29 August, 2015.

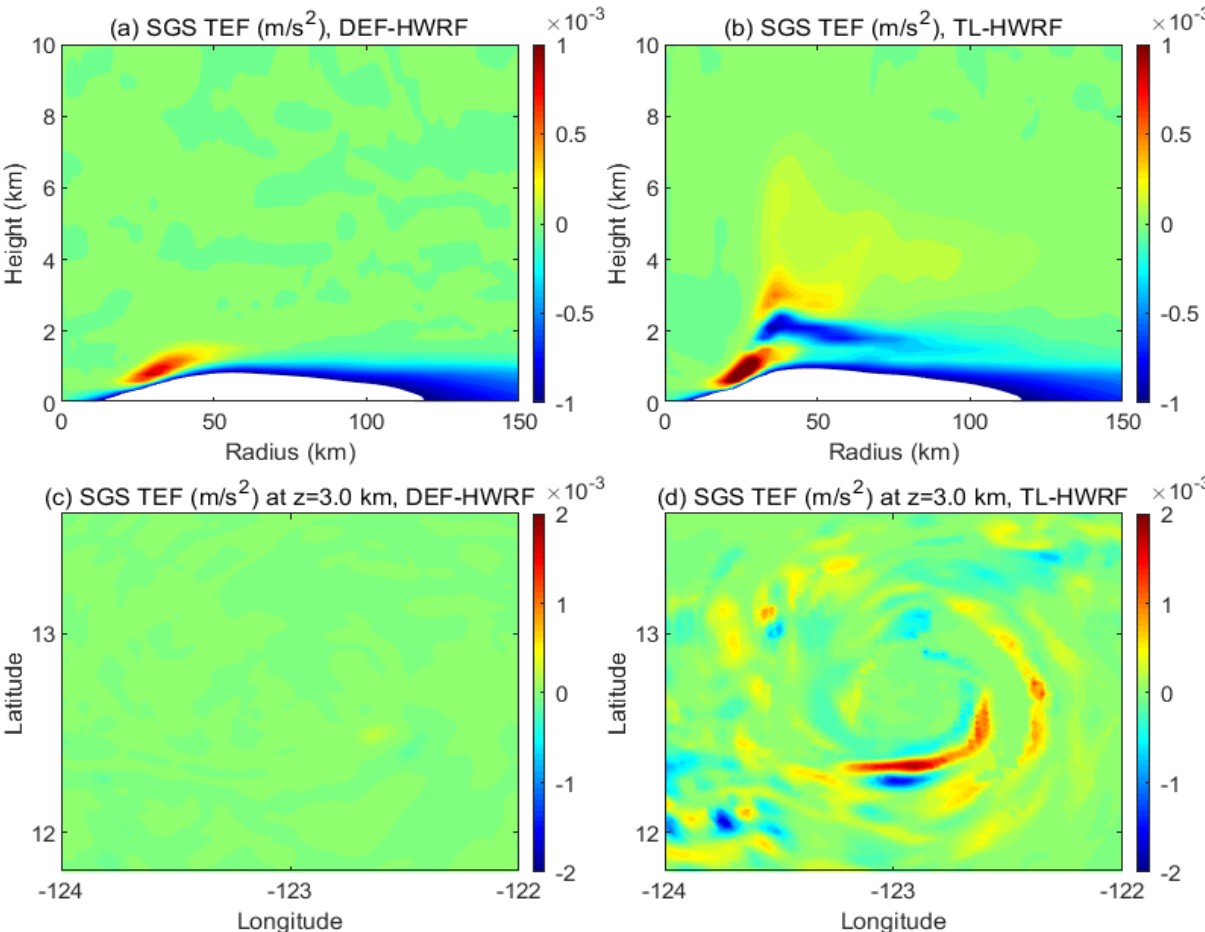

Figure 11: SGS tangential eddy forcing (TEF) averaged over Jimena's RI period from 06:00 UTC 28 to 06: UTC 29 August, 2015 from the two HWRF simulations (DEF-HWRF and TL-HWRF). Top panels: azimuthal-mean radius-height structure of SGS TEF. Note that the SGS TEF smaller than -1.0e-3 (ms$^{-2}$) is shaded with white color for a clear illustration of SGS TEF above the PBL. Bottom panels: horizontal structure of SGS TEF at 3 km altitude.

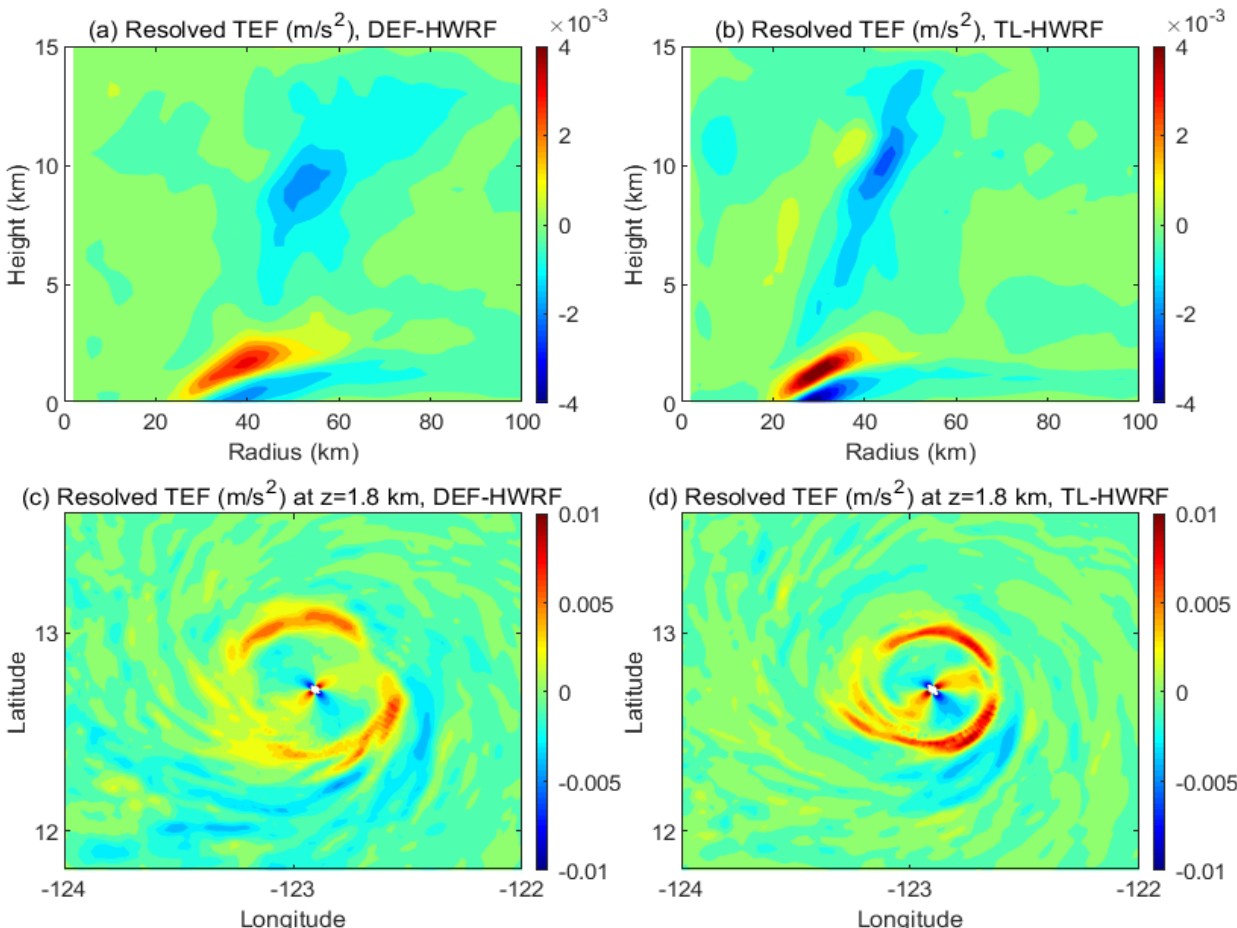

Figure 12: Model-resolved tangential eddy forcing (TEF) averaged over Jimena's RI period from 06:00 UTC 28 to 06:00 UTC 29 August, 2015 from the two HWRF simulations (DEF-HWRF and TL-HWRF). Top panels: azimuthal-mean radius-height structure of resolved TEF. Bottom panels: horizontal structure of resolved TEF at 1.8 km altitude.

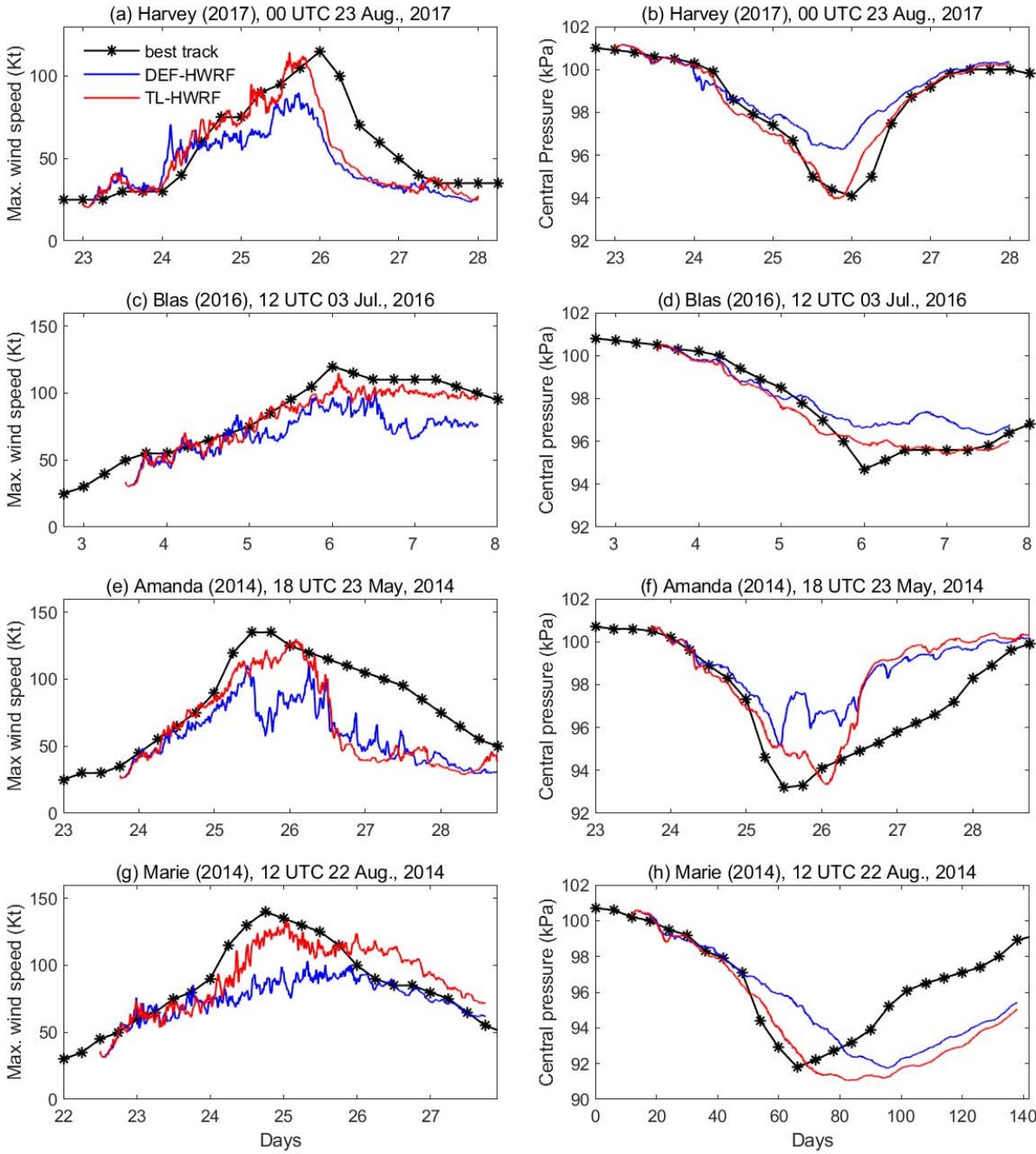

Figure 13: HWRF simulated maximum wind speed and storm central pressure of four other major hurricanes, Harvey (2017), Blas (2016), Amanda (2014), and Marie (2014), compared with the best track data (black curves). Blue curves: DEF-HWRF; Red curves: TL-HWRF.

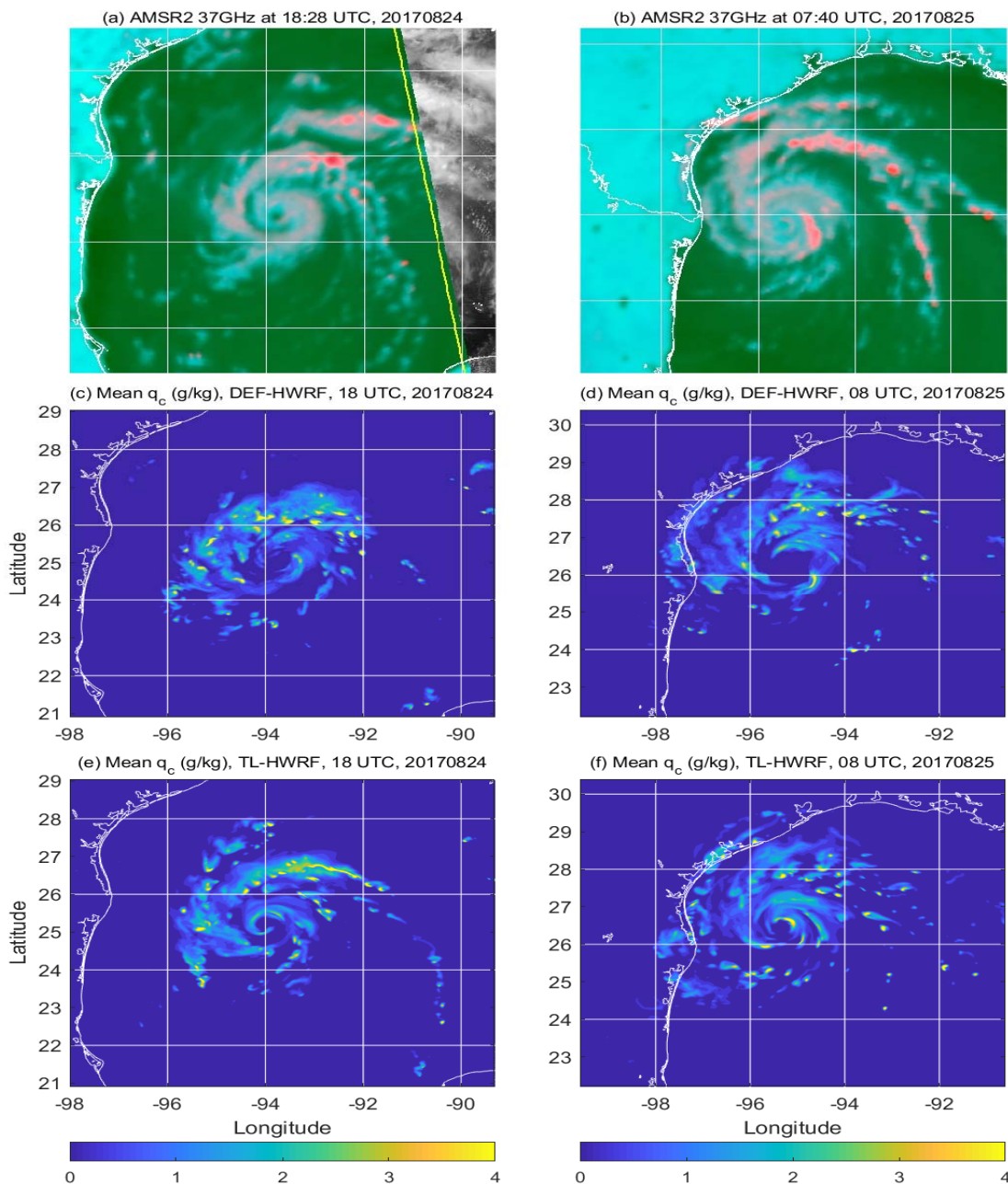

Figure 14: Comparison of vortex inner-core structure during the early stage (left column) and mid stage (right column) of Harvey (2017)'s RI between satellite (AMSR2) observations (top panels) at 18:28 UTC 24 and 07:40 UTC 25 August 2017 and HWRF simulations by DEF-HWRF (middle panels) and TL-HWRF (bottom panels) at 18:00 UTC 24 and 08:00 UTC 25 August 2017. The shown simulated fields are the hydrometeor mixing ratio (gkg$^{-1}$) at 5.0 km altitude.

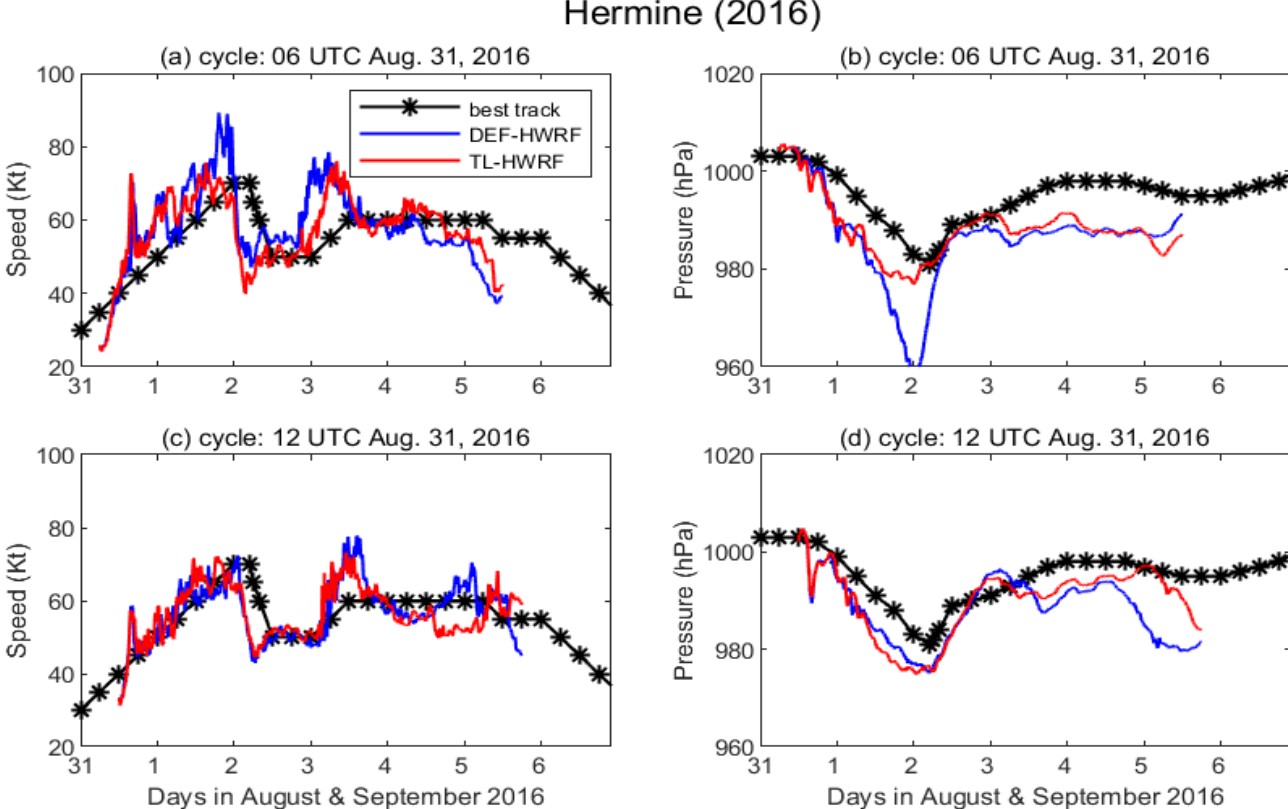

Figure 15: Comparison of HWRF simulated maximum surface wind speed, storm central pressure, and track of Hermine (2016) with the best track data (Black). Blue curves: DEF-HWRF. Red curves: TL-HWRF.

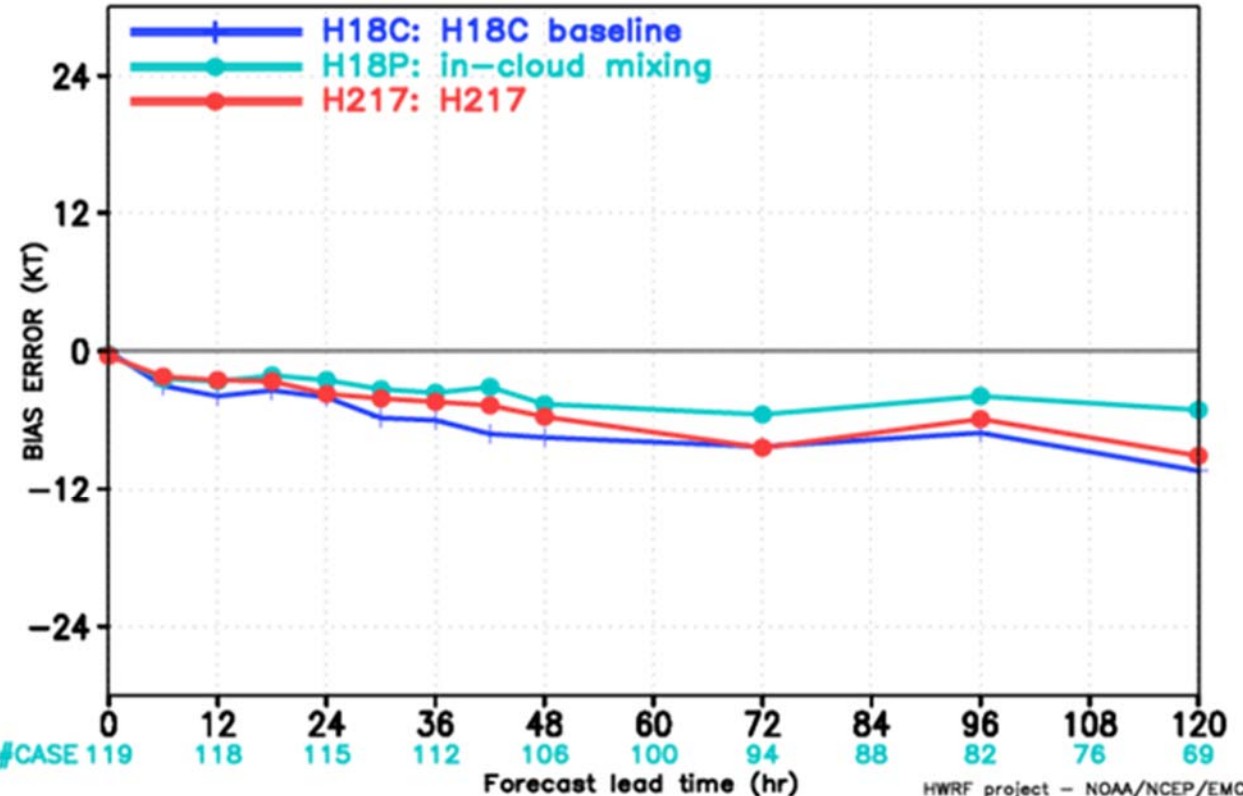

Figure 16: Maximum wind speed bias error (kt) as a function of forecast lead time (hr) averaged over all tested storms and cycles. Average bias errors are shown for the 2018 HWRF model baseline (H18C, blue), 2018 HWRF model with the inclusion an in-cloud turbulent mixing parameterization (H18P, cyan), and 2017 operational HWRF model (H217, red). The storms tested included Hermine (2016), Harvey (2017), Irma (2017), Maria (2017), and Ophelia (2017). The total number of simulation cases for various forecast lead times is indicated by the cyan labels at the bottom (Courtesy to Dr. Sergio Abarca at EMC, NOAA).