# Peer review of "Role of eyewall and rainband eddy forcing in tropical cyclone intensification"

_Atmospheric Chemistry and Physics, 2018_

## Referee Comment (RC1) · DR. Montgomery (Referee) · 15 Nov 2018

Review of: Role of eyewall and rainband eddy forcing in tropical cyclone intensification

by Ping Zhu, Bryce Tyner, Jun A. Zhang, Eric Aligo, Sundararaman Gopalakrishnan, Frank D. Marks, Avichal Mehra, and Vijay Tallapragada

Summary and Evaluation: This is a potentially interesting and potentially useful study for the scientific community devoted to improving our understanding and numerical weather forecasts of damaging tropical cyclones threatening populated coastal communities throughout the world. However, the current manuscript suffers from limitations involving a lack of clarity, poor scholarship in some places and multiple instances of muddled scientific writing. I was very disappointed to discover this state of affairs, given

the large number of co-authors (7), including several senior (& expert) co-authors.

The authors begin their presentation by trying to argue that improvements in the forecast of rapidly intensifying storms will follow: a) once eddy momentum and eddy heat flux processes are properly accounted for in the eyewall and rainband regions; and b) once in-cloud turbulent mixing parameterizations are developed for the eyewall and rainband regions. These are certainly plausible points of view (but see point 2 below for an opposing view). The authors then propose a simple revision of the sub-grid-scale (SGS) turbulence closure scheme for the Hurricane Weather Research Forecast (HWRF) model. The proposed scheme recognizes the prevalence of turbulence in deep (presumably) rotating convection in tropical cyclone vortices. The revised closure is elementary and consists of redefining the height of the boundary layer in deep convective regions, such as the developing eyewall or rainband region, to a height of approximately 5 km altitude based on a model-derived reflectivity threshold of 28 dBZ (Figs. 4e, 4f). This revised boundary layer height definition is called the 'turbulence layer' (TL) and is an extension of the current HWRF scheme (based on pioneering work of Larry Marht and colleagues, and subsequent work in 1996 by Hong and Pan, etc.).

HWRF model simulation experiments invoking the new turbulence parameterization appear to be significantly improved over the standard Hong and Pan (1996) scheme that uses a gradient Richardson-number to define the boundary layer height (typically 1 km). Although the eddy momentum and heat flux divergent tendencies diagnosed from the new simulations are shown to be approximately five times greater than the corresponding SGS tendencies (pg. 13, bottom paragraph, Figs. 10, 11), the authors argue that the revised SGS tendencies are ultimately responsible for the improved forecasts. The authors appear to base their assertion on some mysterious coupling between the turbulence closure scheme and the cloud microphysical processes, and its corresponding coupling to the latent heating rate field associated with the aggregate of deep (presumably) rotating clouds in the inner-core region of the developing

vortex. I am certainly willing to entertain the scientific possibility of a subtle nonlinear feedback involving the SGS tendencies and the microphysics, but the proffered feedback mechanism should be clearly articulated in this manuscript to help support the empirical evidence of the HWRF experiments in real cases. It is unclear, for example, which is most important: the revised turbulence closure scheme for momentum, heat, or moisture?

An alternative (and simpler) hypothesis might be that the structure of the resolved eddy forcing between the control and updated experiments might be more important in accounting for the improved forecasts. This alternate hypothesis originates from a cursory examination of Fig. 11 wherein the resolved eddy forcing of the mean tangential velocity tendency equation in the TL-HWRF experiment is more spatially concentrated and of higher intensity than the DEF-HWRF experiment. (Of course, the different eddy forcings are in part the result from the different SGS formulations, but the larger magnitude of the resolved eddy forcing seems to be a more plausible agent for influencing the spin up process.)

Finally, throughout the manuscript, I was disappointed to find that the authors never asked the basic question of whether a down-gradient turbulence closure for all predicted quantities is indeed appropriate in the rotating, convective turbulence region that pervades a rapidly intensifying tropical cyclone vortex? (see, e.g., Persing et al. 2013, their section 6.)

There are other substantive issues that need to be addressed by the authors and these issues are noted below.

Recommendation: Major Revision. The paper requires substantial improvement in several areas (listed above and below) before I can consider recommending the paper for acceptance in this journal.

Major comments:

[Figure]

First and foremost, the entire manuscript needs to be read carefully by the native English speaking co-authors. I have come across multiple instances of ambiguous or inaccurate statements that need attention. I highlight some of these instances below. I have not provided an exhaustive list however.

1. The first sentence of the Abstract typifies the lack of clarity that occurs in the manuscript:

"The fundamental mechanism underlying tropical cyclone (TC) intensification may be understood from the conservation of absolute angular momentum, where the primary circulation of a TC is driven by the torque acting on air parcels resulting from asymmetric eddy processes, including turbulence."

If the fundamental mechanism underlying TC intensification can be understood from the material conservation of absolute angular momentum (AAM), why, then, are eddy torques being invoked in the SAME sentence to explain how the primary circulation is driven by the torque acting on air parcels resulting from asymmetric eddy processes, including turbulence? While I might be called out for singling out one sentence of the paper, it is the first sentence of the Abstract. Sentences like this abound in the manuscript and portray an alarming state of confusion concerning the mechanisms of tropical cyclone intensification.

2. How come the CHIPS model (Emanuel et al. 2004) is never mentioned in this paper? How come the latest Emanuel (2012) theory for tropical cyclone intensification is never mentioned in this manuscript?

The reason I am asking these questions is that some have suggested that the CHIPS model currently beats all deterministic forecast models (see Jonathan Vigh's talk from the recent AMS conference on Hurricanes and Tropical Meteorology in Ponte Verde, Florida, April 2018). In that context, some have advocated that the turbulence closure problem examined here is a red herring. How will the authors address these questions in the revised manuscript?

3. Pg. 2, Line 11: What is the mechanism underlying TC intensification? Surely, the material conservation of M above the BL (the ice skater model, i.e.) is an essential element of the spin up process above the frictional boundary layer. But what is the mechanism that supports the continued spin up of the vortex? You have not articulated the mechanism(S), other than a passing reference to CISK or cooperative intensification. I believe this is an inadequate state of affairs that needs to be corrected.

4. Pg. 2, lines 19-20: Re eddy forcing in BL:

"In the PBL, eddy forcing ðİŘźðİIJĘ + ðİŘźðİŚăðİŚŤðİŚă_ðİIJĘ is negative definite, meaning that it always slows down the motion; thus, it physically represents the frictional force in the tangential direction. "

". . . is negative definite . . . always slows down the motion" seems to be an assertion without proof of substantiation. Is this a property of the three-dimensional Navier-Stokes equations? (Ans: No.) Is it based in observations? Please give references.

5. Pg. 2, line 29: "In other words, the evolution of the primary circulation of a TC vortex must be (emphasis mine) accompanied by a secondary overturning circulation."

This sentence is insufficiently precise. Purely asymmetric motions can cause an evolution of the mean vortex without mean secondary (overturning) circulation (e.g. a barotropic nondivergent vortex Rossby wave packet and its accompanying wave, mean flow and wave-wave interaction).

Pg. 6, line 1: "Physically, this overturning circulation is induced by (emphasis mine) friction within the PBL and diabatic heating of convection."

Again, this statement should be sharpened. A moving, inviscid, baroclinic vortex on a beta plane will cause asymmetries, which will generally induce an overturning circulation and mean vortex evolution even without friction and diabatic heating (see, e.g., Flatau et al. 1994, JAS).

7. Pg. 3, line 10. Re WISHE and Emanuel 1986. This is a misleading and inaccurate

description of scientific history. Emanuel 1986 is a steady-state hurricane theory (!) and not an intensification theory. The WISHE acronym was not introduced until 5 years later by Yanno and Emanuel 1991. The WISHE feedback mechanism of intensification was articulated by e.g. Emanuel (2003). Following the credible scientific challenges of Montgomery et al. (2009, 2015), WISHE has now been re-defined (Zhang and Emanuel 2016) to mean just the formula for the wind-dependent moist enthalpy flux at the air-sea interface.

Continued:

Line 14. Potentially misleading. Smith and Montgomery were well aware of the limitations of the boundary layer definition used in the hurricane community and noted as such in Smith and Montgomery (2010).

Line 16: Inaccurate. Smith and Montgomery did not assume that the vertical velocity was zero in the boundary layer! (If a slab boundary layer model was being used, then there would be no vertical advection of AAM out of the boundary layer assuming the boundary layer was well mixed in AAM. This is hardly the same as assuming that the vertical velocity is zero within the boundary layer!)

8. Pg. 4, Lines 14-15. Inaccurate. Montgomery and Kallenbach 1997 and Persing et al. 2013 did not root their interpretation of eddy spin up on upscale energy cascade. These authors used a momentum-based approach, which hinges on the eddy vorticity flux (in the barotropic nondivergent case) and the eddy vorticity flux and eddy vertical advection of eddy tangential momentum in the 3D cloud-representing configuration (see Persing et al. 2013, their section 6). The difference is subtle because the eddies can act locally to spin up the maximum mean tangential wind and radial inflow/outflow even while consuming energy from the system-scale mean vortex.

Continued:

Line 26: "multiplication by density first" is missing.

9. Pg. 5, Lines 4-5: "In numerical simulations, Eq. (6) is the equation that governs the azimuthal-mean overturning circulation of a TC vortex."

This statement is physically misleading. The azimuthal mean overturning circulation in a legitimate 3D forecast model such as HWRF is governed in part by the radial and vertical momentum equations. Equation (6) is merely a constraint that must apply at all times and does not "govern" the overturning circulation.

Continued:

Line 6: "In classic TC theories". Citations please.

Lines 7-8: This statement is incorrect. The eddy forcing terms can be zero, but acceleration terms may still be nonzero.

Line 10: Why is Equation (8) to be time differentiated? Please explain.

10. Pg. 7, the text pertaining to Equations (9) and (10). My reading of this text is as follows: the HWRF model uses this two-component formulation for Km (i.e. the Hong and Pan closure in the BL/TL (Eq. 9) and the Smagorinsky closure with stability modification by Lilly above the BL/TL (Eq. 11), respectively) in the vertical and horizontal mixing terms for momentum, heat and moisture. Is this summary correct? Please clarify.

Bryan and Rotunno (2009) and Bryan (2012) use a much higher value of Km for horizontal diffusion than employed here. How do the authors explain this difference in model formulation compared to Bryan and Rotunno?

11. Pg. 14, Lines 1-2. " . . . since the large energy-containing turbulent eddies are not resolved at the current model resolution of 2 km." What are these large energy-containing eddies in these simulations and in real-life tropical cyclones?

12. Figures 10 and 11. The resolved eddy forcing tendency for the azimuthally-averaged tangential velocity tendency equation is plotted in cross section and on several horizontal height surfaces. In these panels the units are displayed as inverse

seconds. Shouldn't the units be that of acceleration (if instantaneous tendencies averaged over some finite time interval) or meters per second (if integrated over some time finite interval)? Please clarify here and elsewhere.

Please also note the supplement to this comment:
https://www.atmos-chem-phys-discuss.net/acp-2018-610/acp-2018-610-RC1-supplement.pdf

---

## Author Comment (AC1) · 8 Jan 2019

Summary and Evaluation: This is a potentially interesting and potentially useful study for the scientific community devoted to improving our understanding and numerical weather forecasts of damaging tropical cyclones threatening populated coastal communities throughout the world. However, the current manuscript suffers from limitations involving a lack of clarity, poor scholarship in some places and multiple instances of muddled scientific writing. I was very disappointed to discover this state of affairs, given the large number of co-authors (7), including several senior (& expert) co-authors.

**We very much appreciate Dr. Montgomery for his thoughtful and constructive comments. The manuscript has been revised accordingly based on his comments. We hope that the revised manuscript is satisfactory for publication. Below is the point-to-point response to reviewer's comments.**

The authors begin their presentation by trying to argue that improvements in the forecast of rapidly intensifying storms will follow: a) once eddy momentum and eddy heat flux processes are properly accounted for in the eyewall and rainband regions; and b) once in-cloud turbulent mixing parameterizations are developed for the eyewall and rainband regions. These are certainly plausible points of view (but see point 2 below for an opposing view). The authors then propose a simple revision of the sub-grid-scale (SGS) turbulence closure scheme for the Hurricane Weather Research Forecast (HWRF) model. The proposed scheme recognizes the prevalence of turbulence in deep (presumably) rotating convection in tropical cyclone vortices. The revised closure is elementary and consists of redefining the height of the boundary layer in deep convective regions, such as the developing eyewall or rainband region, to a height of approximately 5 km altitude based on a model-derived reflectivity threshold of 28 dBZ (Figs. 4e, 4f). This revised boundary layer height definition is called the 'turbulence layer' (TL) and is an extension of the current HWRF scheme (based on pioneering work of Larry Marht and colleagues, and subsequent work in 1996 by Hong and Pan, etc.).

**Two things we want to clarify here. First, we are not saying that the improvements in the numerical forecast of intensification of TCs solely depend on a correct accounting for eddy momentum/heat fluxes and appropriate parameterization of in-cloud turbulent mixing in the eyewall and rainbands. It is a required but not sufficient condition in 3D full-physics simulations. There are important environmental factors, such as SST and large-scale wind shear, and other internal dynamics that can affect TC intensification. In this study, we show that for certain environmental conditions, the numerical simulations of TC intensification, in particular rapid intensification (RI), are sensitive to the parameterization of eyewall/rainband in-cloud turbulent mixing above the boundary layer. Such a sensitivity is largely related to the fact that the parameterized in-cloud turbulent mixing shows a strong bearing on the TC inner-core structure, model-resolved eddy forcing, and secondary overturning circulation.**

**Second, as we admitted in the manuscript (page 10, lines 25-28), the method that we used in this study to include the effects of in-cloud turbulent mixing in HWRF is crude and does not**

consider the specific mechanisms in generating in-cloud turbulence, and thus, the scheme in its current form may not be directly used in operational forecasts. Nonetheless, this simple modification clearly demonstrates the sensitivity of TC intensification to in-cloud turbulent mixing parameterization. It allows us to look into and examine the role of eyewall/rainband eddy forcing above the PBL in TC intensification.

HWRF model simulation experiments invoking the new turbulence parameterization appear to be significantly improved over the standard Hong and Pan (1996) scheme that uses a gradient Richardson-number to define the boundary layer height (typically 1 km). Although the *eddy momentum and heat flux divergent tendencies* diagnosed from the new simulations are shown to be approximately *five times greater than the corresponding SGS tendencies (pg. 13, bottom paragraph, Figs. 10, 11), the authors argue that the revised SGS tendencies are ultimately responsible for the improved forecasts.* The authors appear to base their assertion on some mysterious coupling between the turbulence closure scheme and the cloud microphysical processes, and its corresponding coupling to the latent heating rate field associated with the aggregate of deep (presumably) rotating clouds in the inner-core region of the developing vortex.

Indeed, our diagnoses show that the model-resolved eddy forcing is about five times larger than the SGS eddy forcing above the boundary layer, but the former shows a strong dependence on the parameterization of in-cloud SGS turbulent mixing processes. Such a dependence likely stems from the fact that large energy-containing turbulent eddies are not resolved but parameterized in HWRF. It may also result from the microphysical-dynamical interaction since in-cloud turbulent mixing is intimately involved with cloud microphysics. But the coupling between the parameterized microphysical and turbulent mixing processes is poorly understood. To clarify issues associated with the microphysical-dynamical interaction in 3D full-physics simulations will be one of the focuses of our future research.

I am certainly willing to entertain the scientific possibility of a subtle nonlinear feedback involving the SGS tendencies and the microphysics, but the proffered feedback mechanism should be clearly articulated in this manuscript to help support the empirical evidence of the HWRF experiments in real cases. It is unclear, for example, which is most important: the revised turbulence closure scheme for momentum, heat, or moisture?

We have done two additional experiments. In the first experiment, we only modified the eddy exchange coefficient for momentum $K_m$ while keeping the eddy exchange coefficient for heat and moisture $K_{t,q}$ the same as the default. We reversed such a change in the second experiment. Figure R1 below compares the simulated intensity and track of Hurricane Jimena (2015) from different numerical experiments. Both the modified turbulence closures for momentum alone and for heat/moisture alone show non-negligible impacts on TC intensification. This result is not unexpected. While the tangential eddy forcing for momentum directly involves in the acceleration or deceleration of the primary circulation of a TC, the thermodynamic eddy forcing is sufficiently strong to modulate the secondary

overturning circulation that interacts with the primary circulation during TC evolution. Note that in numerical models including HWRF, the eddy exchange coefficients for heat and moisture are usually treated as the same, thus, we did not separate them in our experiments.

These two addition experiments have been included in the revised manuscript. Please see Page 12, lines 11 – 20, and updated Figure 6 in the revised manuscript.

[Figure]

Figure R1: Comparison of HWRF simulated maximum surface wind speed, storm central pressure, and track of Jimena (2015) with the best track data (Black). Blue curve indicates the simulation by the default HWRF (DEF-HWRF). Red curve indicates the simulation by the HWRF with inclusion of in-cloud turbulent mixing parameterization (TL-HWRF). Green curve represents the simulation in which only the eddy exchange coefficient for momentum is modified while keeping the eddy exchange coefficient for heat and moisture the same as the default. Magenta curve is opposite to the green curve in which only the eddy exchange coefficient for heat and moisture is modified.

An alternative (and simpler) hypothesis might be that the structure of the resolved eddy forcing between the control and updated experiments might be more important in accounting for the improved forecasts. This alternate hypothesis originates from a cursory examination of Fig. 11 wherein the resolved eddy forcing of the mean tangential velocity tendency equation in the TL-HWRF experiment is more spatially concentrated and of higher intensity than the DEF-HWRF experiment. (Of course, the different eddy forcings are in part the result from the different SGS formulations, but the larger magnitude of the resolved eddy forcing seems to be a more plausible agent for influencing the spin up process.)

We agree with the comments. Indeed, the TC inner-core structure shows a substantial sensitivity to the in-cloud turbulent mixing parameterization in the eyewall and rainbands. The larger magnitude of the resolved eddy forcing appears to be a more plausible agent for the spin-up process. We will further explore this sensitivity in our future research.

Finally, throughout the manuscript, I was disappointed to find that the authors never asked the basic question of whether a *down-gradient turbulence closure for all predicted quantities* is indeed appropriate in the rotating, convective turbulence

region that pervades a rapidly intensifying tropical cyclone vortex? (see, e.g., Persing et al. 2013, their section 6.)

How to parameterize SGS turbulence is not only critical for predicting TCs but important for simulating all other atmospheric phenomena as well. But depending on the problems, the focuses of turbulent mixing parameterization could be different. For example, for shallow stratocumulus clouds, the above boundary layer turbulence is negligible since the clouds are normally capped by a strong inversion. Because of that, representing turbulent mixing associated with the strong cloud-top entrainment becomes important to successfully simulate the evolution of stratocumulus-topped boundary layer. In this case, an appropriate treatment of cloud-top radiave cooling and evaporative cooling is a key since they are intimately involved in the buoyancy production of turbulence in clouds. Drizzling is also an important factor to consider since it can cause decoupling of the cloudy boundary layer. In a TC environment, however, the boundary layer top entrainment is a minor since no inversion can be found in convective regimes except for the TC eye or moat regions where weak inversion may exist. But in this case, the above-boundary-layer turbulent mixing generated by the cloud processes in the eyewall and rainbands becomes important since the in-cloud buoyancy-driven turbulent eddies can generate efficient transport in the vertical and can directly interact with cloud microphysics. Although our study does touch the problem of how to parameterize in-cloud turbulence, the main issue that we are addressing in this paper is what needs to be included in the turbulent mixing scheme for a successful numerical prediction of TCs. We believe that the in-cloud turbulent mixing above the boundary layer in the eyewall and rainbands is a key component of SGS turbulent processes that must be considered in TC simulations. But it has not been included or well represented in current models used for TC prediction.

As for the down-gradient turbulence closure, it is a commonly used approach for representing turbulent transport to mimic molecular diffusion. The difference is that the conductivity for down-gradient molecular diffusion is determined by material, but the eddy exchange coefficient for down-gradient turbulent transport is function of motion (or turbulence). But the down-gradient transport is basically a local transport mechanism, it does not hold for convective cells or large eddies because their transport mechanism is fundamentally nonlocal. This is a problem has been recognized for a long time. Nowadays, many turbulent mixing schemes do include nonlocal mixing component when parameterizing vertical turbulent transport. For example, YSU, NCEP Global Forecast System scheme, ACM2, UW PBL scheme, TEMF, Shin-Hong Scale-aware scheme, Grenier-Bretherton-McCaa scheme, MRF, and others all include non-local mixing effect in their schemes. The PBL scheme (or vertical turbulent mixing scheme) used in the latest version of HWRF also considers non-local mixing based on the GFS PBL scheme. This is different from the horizontal mixing schemes used in models. To our knowledge, none of them considers non-local mixing in horizontal turbulent mixing parameterization. Most of models use Smagorinsky type (local) schemes to treat SGS horizontal turbulent mixing. As we stated previously, the main focus of this paper is not to discuss the advantage and disadvantage of down-gradient (local) and non-local mixing approaches, but to address if in-cloud turbulent mixing generated by cloud processes above the boundary layer in the eyewall and rainbands needs to be included in the vertical turbulent mixing scheme for TC simulations.

There are other substantive issues that need to be addressed by the authors and these issues are noted below.

Recommendation: Major Revision. The paper requires substantial improvement in several areas (listed above and below) before I can consider recommending the paper for acceptance in this journal.

Major comments:

First and foremost, the entire manuscript needs to be read carefully by the native English speaking co-authors. I have come across multiple instances of ambiguous or inaccurate statements that need attention. I highlight some of these instances below. I have not provided an exhaustive list however.

1. The first sentence of the Abstract typifies the lack of clarity that occurs in the manuscript:

"The fundamental mechanism underlying tropical cyclone (TC) intensification may be understood from the conservation of absolute angular momentum, where the primary circulation of a TC is driven by the torque acting on air parcels resulting from asymmetric eddy processes, including turbulence."
If the fundamental mechanism underlying TC intensification can be understood from the material conservation of absolute angular momentum (AAM), why, then, are *eddy torques* being invoked in the SAME sentence to explain *how the primary circulation is driven by the torque* acting on air parcels resulting from *asymmetric eddy processes, including turbulence*? While I might be called out for singling out one sentence of the paper, it is the first sentence of the Abstract. Sentences like this abound in the manuscript and portray an alarming state of confusion concerning the mechanisms of tropical cyclone intensification.

The sentence was meant to describe the budget equation of azimuthal-mean absolute angular momentum (AAM) , $\frac{D\bar{\bar{M}}}{Dt} = r(F_\lambda + F_{sgs\_\lambda})$, where $F_\lambda$ and $F_{sgs\_\lambda}$ are the azimuthal-mean tangential eddy correlation terms resulting from the model-resolved and parameterized sub-grid scale (SGS) asymmetric eddy processes. It is clear that for certain external conditions, such as large-scale wind shear and SST, the spin-up and spin-down of the mean vortex of a TC is driven by the eddy forcing represented by the terms on the right-hand side of the equation. In real TCs, the eddy forcing possesses a continuous spectrum from eyewall/rainband mesoscale convective features down to small-scale turbulence. In numerical simulations, however, the continuous eddy forcing is artificially split into two parts: the model-resolved and parameterized SGS eddy forcing due to discrete model grids. The two split components of eddy forcing not only are sensitive to model grid spacing but also depend strongly on each other. For SGS eddy forcing, the research to date mainly focuses on the boundary layer turbulence. However, the boundary layer classically viewed as a shallow layer adjacent to Earth's surface becomes ill-defined as radial inflow ascends swiftly

along the eyewall. In this case, there is no physical interface that separates the turbulence generated by the shear and buoyancy production associated with the surface processes and by the cloud processes aloft. While the importance of boundary layer turbulent transport to TC intensification is well recognized, little attention has been paid to the turbulent mixing aloft in the eyewall/rainband clouds. The main motivation of this study is to examine the role of eddy forcing resulting from eyewall/rainband clouds above the boundary layer in TC intensification. We showed that the simulated TC inner-core structure and the model-resolved eddy forcing (and thus, the TC intensification) depend strongly on parameterization of in-cloud turbulent mixing.

To avoid confusion, the sentence has been removed in the revised manuscript. The abstract has been rewritten.

2. How come the CHIPS model (Emanuel et al. 2004) is never mentioned in this paper? How come the latest Emanuel (2012) theory for tropical cyclone intensification is never mentioned in this manuscript?

The reason I am asking these questions is that some have suggested that the CHIPS model currently beats *all* deterministic forecast models (see Jonathan Vigh's talk from the recent AMS conference on Hurricanes and Tropical Meteorology in Ponte Verde, Florida, April 2018). In that context, some have advocated that the turbulence closure problem examined here is a red herring. How will the authors address these questions in the revised manuscript?

We are aware of Dr. Jonathan Vigh et al.'s recent work on exploring the upper bound of TC intensification by comparing the rapid intensifications (RIs) simulated by the CHIPS model with those by the state-of-the-art 3D full-physics numerical models (presented at the AMS 33$^{rd}$ Conference on Hurricanes & Tropical Meteorology). The CHIPS model here refers to a simple axisymmetric 2D radius-height dynamic model developed by Emanuel et al. (2004). They showed that CHP6 (an ensemble member of the CHIPS simulations) outperforms all 3D numerical models in terms of guidance for estimating the upper bound of TC intensification. Based on this result, they concluded (their presentation PPT file is available at AMS website) that (1) the dynamics of very RI (VRI ~30 kt in 12 h) and extreme RI (ERI ~ 40 kt in 12 h) are primarily axisymmetric and do not require a 3D full-physics framework; and (2) the general pathway to VRI/ERI can be captured by an axisymmetric numerical model. It should be noted here that CHP6 was initialized with an intensity enhanced by 3 ms$^{-1}$ of the previous 24-hr forecast with vertical wind shear set to ZERO at all times, so, the setting is basically assumed to be ideal for TC intensification. Here we want to clarify several things as follows.

First, we agree with Vigh et al.'s argument that the dynamics governing VRI/ERI are primarily axisymmetric. This result is supported by previous studies. For example, Nolan and Grasso (2003) and Nolan et al. (2007) showed that it is the vortex axisymmetric response to the azimuthally-averaged diabatic heating (rather than the heating associated with individual asymmetries) that is responsible for the resultant intensity change. In fact, our HWRF simulations are consistent with these results. This is clearly seen in Figs. 8b & 8d and Figs. 13e & 13f, the HWRF with the inclusion of in-cloud turbulent mixing parameterization

successfully simulated the nearly axisymmetric ring of convection around the storm center consistent with satellite observations (Fig. 7 and Figs. 13a & 13b). The closed convective ring feature is an evidence to support that the RI of Hurricanes Jimena (2015) and Harvey (2017) indeed can be explained by the axisymmetric dynamics. So, there is no conflict between our simulations and Vigh et al.'s conclusion.

Second, we do not think that the turbulent closure problem investigated in this paper is a red herring to the RI driven by axisymmetric dynamics. The fact that the default HWRF fails to simulate the observed TC inner-core structure including the closed ring feature suggests that the realization of axisymmetric dynamics underlying TC intensification in 3D full-physics models is not scientifically trivial. It is a difficult problem since it involves a complicated interplay between model dynamic core and various physics modules in 3D models. As we showed in the paper, although the direct eddy forcing from the parameterized eyewall/rainband in-cloud turbulent processes is only minor compared with the model-resolved eddy forcing, the simulated TC inner-core structure, the secondary circulations, and model-resolved eddy processes all show substantial sensitivity to the in-cloud turbulent mixing parameterization. Therefore, the impact of SGS processes on TC intensification should not be considered as an isolated effect since it can induce changes in TC internal dynamics. Such a sensitivity of TC evolution to in-cloud turbulent mixing parameterization may stem from the fact that current model grid spacing (~1 – 2 km) used for TC prediction cannot resolve large energy-containing turbulent eddies. It also suggests that the SGS physics involving with the in-cloud turbulent mixing above the PBL facilitates the realization of the axisymmetric dynamics underlying the RI of TCs in 3D full-physics simulations.

Third, there is no doubt that some observed RIs can be well explained by the axisymmetric dynamics, but that does not mean that 3D full-physics models can be replaced by axisymmetric models such as CHIPS for real-time TC prediction. According to Dr. Vigh's presentation, only one ensemble number of CHIPS, CHP6, is able to capture the observed RIs. Figure R2 below shows an example of CHIPS ensemble simulations of Hurricane Patricia (2015). Except for CHP6, all the other member simulations fail to reproduce the observed RI and peak intensity. This is the same for Typhoon Meranti (2016) and Hurricane Maria (2017, not shown here but is available in Dr. Vigh's presentation). Recall that CHP6 was configured ideal for TC intensification with vertical wind shear set to ZERO at all times for all tested TCs. But this is not a realistic ambient condition for some RI TCs. As shown in Fig. R2, when the ambient conditions were set to those used by deterministic forecast modes, the control run of CHIPS ensemble simulations (member CHIP) completely missed the Patricia's RI. So, it is not an accurate statement that CHIPS model "*beats all deterministic forecast models*", it's just one member of CHIPS with ZERO wind shear that generates the intensification rate close to observations in some cases. CHIPS provides a great framework to examine the sensitivity of TC intensification to ambient conditions, but it would be inappropriate to use the ZERO-wind-shear setting for real-time forecasts as the real condition may be far away from it.

Finally, we agree with the argument that the upper limit on intensification is determined by the conversion of latent heating to kinetic energy. The reason that this conversion is often

less efficient in a 3D intensification process than that in a 2D axisymmetric model is likely due to the fact that the convection in a 3D model has not yet organized into an annular ring, and thus, the azimuthally averaged heating rate is smaller than that in a 2D axisymmetric model. This probably is the case in Vigh et al.'s simulations. However, there are examples in literature that showed the energy conversion in 3D model simulations can be as efficient as that in an axisymmetric model. For instance, Persing et al. (2013) reported that "there is a short period of time when the rate of spin-up in the 3-D model exceeds that of the maximum spin-up rate in the axisymmetric model, and during this period the convection is locally more intense than in the axisymmetric model and the convection is organized in a quasi ring-like structure resembling a developing eyewall" (see Page 12336 of their paper). This is a situation similar to our simulations with the inclusion of eyewall/rainband in-cloud turbulent mixing parameterization.

[Figure]

Figure R2: CHIPS ensemble simulations of Hurricane Patricia (2015) adopted from Vigh et al.'s presentation at the AMS 33rd Conference on Hurricanes & Tropical Meteorology.

In 3D numerical simulations, the conversion from latent heating to kinetic energy depends on many factors, such as, model resolution, initial vortex structure, and interaction between resolved and parameterized processes. The representation of SGS physics is apparently a source of uncertainty in modeling TC intensification. In this paper, we mainly focused on one particular problem of SGS physics --- the parameterization of turbulent mixing above the PBL

generated by cloud processes. We have been trying to understand the impact of eyewall/rainband eddy processes (both resolved and SGS) on TC intensification and improve in-cloud turbulent mixing parameterization in operational models.

As for Emanuel (2012) paper, his "self-stratification" intensification hypothesis argued that the turbulence in the outflow is important because it acts to set the thermal stratification of the outflow. The resultant gradients of outflow temperature provide a control of an intensifying vortex. In our study, the defined "Turbulent Layer (TL)" does not include the turbulence generated by the anvil clouds in the upper troposphere where the eyewall upflow turns outward, becoming outflow. Outside a convection regime, the anvil clouds are detached from the PBL in vertical model columns, thus, "TL" concept does not apply. In their analyses (Emanuel and Rotunno 2011; Emanuel 2012), the instability for generating small-scale mixing was estimated by the Richardson number. However, since numerical models use stretching grids in the vertical, it is very difficult to parameterize the SGS turbulent mixing in the outflow regions using bulk Richardson number at a very low vertical resolution. Moreover, since the main focus of this study is on the turbulent mixing generated by cloud processes above the PBL within the eyewall and rainbands, we want to isolate this problem from the complication of outflow turbulence. For these reasons, the outflow turbulence above the PBL is not discussed in this study.

These issues have been clarified in the revised manuscript. Please see Page 6, lines 15 – 27; Page 11, lines 1 – 11; and Page 13, lines 11 – 18 in the revised manuscript.

3. Pg. 2, Line 11: What is the mechanism underlying TC intensification? Surely, the material conservation of M above the BL (the ice skater model, i.e.) is an essential element of the spin up process above the frictional boundary layer. But what is the mechanism that supports the continued spin up of the vortex? You have not articulated the mechanism(S), other than a passing reference to CISK or cooperative intensification. I believe this is an inadequate state of affairs that needs to be corrected.

For an axisymmetric vortex free of forcing in an inviscid flow, the conservation of absolute angular momentum is an essential element of the spin up process of the vortex. But in reality, there are asymmetric eddies with a spectrum of scales from mesoscale convective elements down to small-scale turbulence superimposed on the mean axisymmetric vortex. On a radius-height plane of the mean vortex, the aggregate effects of the asymmetric eddies are represented by the forcing terms appearing in the azimuthal-mean governing equations. For example, the budget equations for azimuthal-mean tangential, raidial, and vertical velocities may be written as,

$$\frac{\partial \bar{v}}{\partial t} + \bar{u}\frac{\partial \bar{v}}{\partial r} + \bar{w}\frac{\partial \bar{v}}{\partial z} + \bar{u}\left(f + \frac{\bar{v}}{r}\right) = F_\lambda + F_{sgs\_\lambda},$$

$$\frac{\partial \bar{u}}{\partial t} + \bar{u}\frac{\partial \bar{u}}{\partial r} + \bar{w}\frac{\partial \bar{u}}{\partial z} - \left(\frac{\bar{v}^2}{r} + f\bar{v}\right) + \frac{1}{\bar{\rho}}\frac{\partial \bar{p}}{\partial r} = F_r + F_{sgs\_r},$$

$$\frac{\partial \bar{w}}{\partial t} + \bar{u}\frac{\partial \bar{w}}{\partial r} + \bar{w}\frac{\partial \bar{w}}{\partial z} + \left(\frac{1}{\bar{\rho}}\frac{\partial \bar{p}}{\partial z} + g\right) = F_w + F_{sgs\_w}.$$

To date, a great effort has been devoted to elucidating how eddy forcing drives the primary and secondary circulations of a TC vortex. The SGS eddy forcing ($F_{sgs\_\lambda}$, $F_{sgs\_r}$, and $F_{sgs\_w}$) results from SGS turbulence, which is commonly regarded as a flow feature in the PBL. The importance of PBL turbulent transport to TC evolution has long been recognized. The early theories of TCs (e.g., Charney and Eliassen 1964, Ooyama 1982, Emanuel 1986, Emanuel 2003) all recognized the role of turbulence in transporting latent heating obtained from ocean surface and converging moisture in the PBL to sustain eyewall/rainaband deep convection. From eddy forcing perspective, all these studies basically focused on the SGS eddy forcing within the PBL since the PBL turbulent transport is a parameterized quantity in their models. But the role of eddy forcing resulting from asymmetric convective elements in TC evolution cannot be addressed by these studies since axisymmetric vortices were used. The advanced 3D rotating convective updraft paradigm (Montgomery and Smith 2014) recognized the importance of asymmetries, such as hot towers, to TC intensification. Persing et al. (2013) compared the TC intensification rate in a 3D full-physics model with that in an axisymmetric model.

What has received little attention and has yet be explored is the SGS eddy forcing above the boundary layer. As we stated in the paper, the classic definition of boundary layer does not apply to the eyewall and rainbands because there is no physical interface separating the turbulence generated by the buoyancy and shear production related to the surface processes and cloud processes aloft. The turbulent processes associated with the eyewall/rainband convective clouds also acquire nearly annulus-like or spiral feature depending on the detailed structures of eyewall and rainbands. The in-cloud turbulent mixing is thus important not only because it forms a component of eddy forcing directly involving in the evolution of the primary and secondary circulations of a TC but also because it interacts with cloud microphysics to affect diabatic heating. This may explain why the simulated TC inner-core structure, secondary circulation, and model-resolved eddy forcing are all sensitive to in-cloud turbulent mixing parameterization. Moreover, since numerical models usually have a coarse vertical resolution above the boundary layer, it makes difficult to appropriately parameterize in-cloud turbulent mixing in models.

We have redrafted the related paragraphs and sentences in the revised manuscript based on the comments. Please see Page 2, lines 23 – 28; Page 3, lines 1 – 19; Page 4, lines 13 – 28; and Page 5, lines 4 – 14 in the revised manuscript.

4. Pg. 2, lines 19-20: Re eddy forcing in BL:

"In the PBL, eddy forcing $F\lambda$ + $Fsgs\_\lambda$ is negative definite, meaning that it always slows down the motion; thus, it physically represents the frictional force in the tangential direction. "

"... is negative definite ... always slows down the motion" seems to be an assertion without proof of substantiation. Is this a property of the three-dimensional Navier-Stokes equations? (Ans: No.) Is it based in observations? Please give references.

This statement was written following Montgomery and Smith (2014) paper (*Paradigms for Tropical Cyclone Intensification*). On Page 41-42 in their paper, they wrote: "The key element of vortex spin-up in an axisymmetric setting can be illustrated from the equation for absolute angular momentum per unit mass, $\frac{\partial M}{\partial t} + u\frac{\partial M}{\partial r} + w\frac{\partial M}{\partial z} = F$. ... In regions where frictional forces are appreciable, $F$ is negative definite (provided that the tangential flow is cyclonic relative to the earth's local angular rotation, $v/r > -f$), and $M$ decreases following air parcels.". We interpreted that "the regions where frictional forces are appreciable" refer to the boundary layer. Negative $F$ in the friction layer is consistent with what Charney and Eliassen (1964) stated that friction acts to dissipate kinetic energy, and is also supported by Fig. 10 and Fig. 11 in our paper where we showed the eddy forcing is negative in the inflow layer.

The reference has been added in the revised manuscript, and the related sentence has been rewritten. Please see Page 2, lines 14 – 15 in the revised manuscript.

5. Pg. 2, line 29: "In other words, the evolution of the primary circulation of a TC vortex *must be* (emphasis mine) accompanied by a secondary overturning circulation."

This sentence is insufficiently precise. Purely asymmetric motions can cause an evolution of the mean vortex without mean secondary (overturning) circulation (e.g. a barotropic nondivergent vortex Rossby wave packet and its accompanying wave, mean flow and wave-wave interaction).

**Agree. Indeed, as shown by Montgomery and Kallenbach (1997), the propagation of vortex Rossby wave packets and the associated wave-mean-flow interaction can lead to the evolution of vorticity monopoles without the mean secondary circulation in a barotropic nondivergent framework.**

**The sentence has been removed from the revised manuscript. The paragraph has been rewritten.**

Pg. 6, line 1: "Physically, this overturning circulation *is induced by* (emphasis mine) *friction* within the PBL and *diabatic heating* of convection."
Again, this statement should be sharpened. A moving, inviscid, baroclinic vortex on a beta plane will cause asymmetries, which will generally induce an overturning circulation and mean vortex evolution even without friction and diabatic heating (see, e.g., Flatau et al. 1994, JAS).

**Agree. For a baroclinic vortex, the horizontal advection of the relative vorticity caused by the β effect is much greater in the lower troposphere (where the vortex is strongest) than it is in the upper troposphere (where the vortex is weakest). This causes the lower tropospheric portion of the vortex to be advected to the northwest more rapidly than it is in the upper troposphere. It also results in a slightly upwind-tilted structure of the vortex. The quasi-geostrophic omega equation enables us to diagnose vertical motions as a function of the**

differential advection of cyclonic relative vorticity. In this scenario, forcing for descent is found to the northwest and forcing for ascent is found to the southeast. We note that the resulting secondary (overturning) circulation by the β effect is much weaker than that induced by friction in the boundary layer and diabatic heating.

**The sentence has been removed from the revised manuscript. The paragraph has been rewritten.**

7. Pg. 3, line 10. Re WISHE and Emanuel 1986. This is a misleading and inaccurate description of scientific history. Emanuel 1986 is a steady-state hurricane theory (!) and not an intensification theory. The WISHE acronym was not introduced until 5 years later by Yanno and Emanuel 1991. The WISHE *feedback mechanism* of intensification was articulated by e.g. Emanuel (2003). Following the credible scientific challenges of Montgomery et al. (2009, 2015), WISHE has now been redefined (Zhang and Emanuel 2016) to mean just the formula for the wind dependent moist enthalpy flux at the air-sea interface.

**Agree. The sentence has been removed from the revised manuscript. The paragraph has been rewritten.**

Continued:

Line 14. Potentially misleading. Smith and Montgomery were well aware of the limitations of the boundary layer definition used in the hurricane community and noted as such in Smith and Montgomery (2010).

**Smith and Montgomery (2010) has been referenced in the paragraph starting with line 25. This is one of papers that motivated our study.**

Line 16: Inaccurate. Smith and Montgomery did not assume that the vertical velocity was zero in the boundary layer! (If a slab boundary layer model was being used, then there would be no vertical advection of AAM *out of the boundary layer* assuming the boundary layer was well mixed in AAM. This is hardly the same as assuming that the vertical velocity is zero *within* the boundary layer!)

**We agree on the comments that Smith and Montgomery did not assume the vertical velocity was zero in the boundary layer. However, there is something we want clarify here. It is important to distinguish the mean vertical velocity $\bar{w}$ and vertical velocity fluctuations $w'$ in the boundary layer. Although $\bar{w}$ is important in mass conservation and in the advection of material (e.g., moisture), it is less important in terms of vertical fluxes since it is always paired in a linear manner with $w'$. In the boundary layer, $\bar{w}$ is considerably small compared to $w'$. When calculating vertical turbulent fluxes, $\bar{w}$ is often neglected (Stull 1988). This is what we meant when we say "the mean vertical velocity in the PBL is negligible" in Line 16.**

**This has been clarified in the revised manuscript. Please see Page 3, lines 23 – 24 in the revised manuscript.**

8. Pg. 4, Lines 14-15. Inaccurate. Montgomery and Kallenbach 1997 and Persing et al. 2013 did not root their interpretation of eddy spin up on upscale energy cascade. These authors used a momentum-based approach, which hinges on the eddy vorticity flux (in the barotropic nondivergent case) and the eddy vorticity flux and eddy vertical advection of eddy tangential momentum in the 3D cloud-representing configuration (see Persing et al. 2013, their section 6). The difference is subtle because the eddies can act *locally* to spin up the maximum mean tangential wind and radial inflow/outflow *even while consuming energy from the system-scale mean vortex.*

**Agree. The sentence has been removed from the revised manuscript. The paragraph has been rewritten.**

Continued:

Line 26: "multiplication by density first" is missing.

**Agree. The paragraph has been rewritten in the revised manuscript.**

9. Pg. 5, Lines 4-5: "In numerical simulations, Eq. (6) is the equation that governs the azimuthal-mean overturning circulation of a TC vortex."

This statement is physically misleading. The azimuthal mean overturning circulation in a legitimate 3D forecast model such as HWRF is governed in part by the radial and vertical momentum equations. Equation (6) is merely a constraint that must apply at all times and does not "govern" the overturning circulation.

**Agree. In 3D models, the azimuthal-mean overturning circulation is governed in part by the radial and vertical momentum equations.**

**However, one may derive a secondary overturning circulation from Eq. (6) with certain assumptions. For example, neglecting eddy forcing terms in Eq. (6) and assuming that a vortex is in a steady state, i.e., a vortex satisfying hydrostatic balance and gradient wind balance, Eq. (6) simplifies to the thermal wind relationship, $g\frac{\partial \overline{x}}{\partial r} + \frac{\partial (\overline{x}C)}{\partial z} = 0$, where $\overline{\chi} = \frac{1}{\overline{\theta}}$. Smith et al. (2005) and Bui et al. (2009) showed that by taking the time derivative of this equation, eliminating the time derivatives of $\overline{x}$ and $\overline{v}$ using the azimuthal-mean heat budget equation tangential wind budget equation, and representing the azimuthal-mean radial and vertical velocity in terms of a streamfunction ($\overline{u} = -\frac{1}{r\overline{\rho}}\frac{\partial \overline{\psi}}{\partial z}, \overline{w} = \frac{1}{r\overline{\rho}}\frac{\partial \overline{\psi}}{\partial r}$), an analytical expression of the overturning circulation known as Sawyer-Eliassen equation (SEE) can be derived,**

$$\frac{\partial}{\partial r}\left[-\frac{1}{r\overline{\rho}}\frac{\partial \overline{\psi}}{\partial z}\frac{\partial (C\overline{\chi})}{\partial z} - g\frac{1}{r\overline{\rho}}\frac{\partial \overline{\psi}}{\partial r}\frac{\partial \overline{\chi}}{\partial z}\right] + \frac{\partial}{\partial z}\left[\left(\overline{\chi}\xi(f+\zeta) + C\frac{\partial \overline{\chi}}{\partial r}\right)\frac{1}{r\overline{\rho}}\frac{\partial \overline{\psi}}{\partial z} - \frac{1}{r\overline{\rho}}\frac{\partial \overline{\psi}}{\partial r}\frac{\partial (C\overline{\chi})}{\partial z}\right] = g\frac{\partial}{\partial r}(\overline{\chi}^2 Q) +$$
$$\frac{\partial}{\partial z}(C\overline{\chi}^2 Q) - \frac{\partial}{\partial z}\left[\overline{\chi}\xi(F_\lambda + F_{sgs_\lambda})\right],$$

where $\xi = f + \frac{2\bar{v}}{r}$, $\zeta = \frac{\bar{v}}{r} + \frac{\partial \bar{v}}{\partial r}$, $C = \frac{\bar{v}^2}{r} + f\bar{v}$, and $Q = \dot{\theta} + F_\theta + F_{sgs\_\theta}$ representing diabatic heating and eddy forcing for heat. Since the hydrostatic balance and gradient wind balance are used, SEE can only diagnose the overturning circulation in response to diabatic heating and tangential eddy forcing. It does not provide any information on how radial and vertical eddy forcing affects the overturning circulation. Smith et al. (2009) showed that one of the spin-up mechanisms of the mean tangential circulation involves the convergence of AAM within the boundary layer associated with the development of supergradient wind speeds in the boundary layer.

With Eq. (6), one may derive a SEE-like equation to diagnose the mean overturning circulation in response to those factors that were omitted by SEE. For example, we may include the radial eddy forcing in the simplified Eq. (6), then, the thermal wind relationship is replaced by $g\frac{\partial \bar{\chi}}{\partial r} + \frac{\partial [\bar{\chi}(C+F_R)]}{\partial z} = 0$, where $F_R = F_r + F_{sgs\_r}$ is the radial eddy forcing. Following the same procedure, one may derive a SEE-like equation,

$$\frac{\partial}{\partial r}\left[-\frac{1}{r\bar{\rho}}\frac{\partial \bar{\psi}}{\partial z}\frac{\partial(C\bar{\chi})}{\partial z} - g\frac{1}{r\bar{\rho}}\frac{\partial \bar{\psi}}{\partial r}\frac{\partial \bar{\chi}}{\partial z}\right] + \frac{\partial}{\partial z}\left[\left(\bar{\chi}\xi(f+\zeta) + C\frac{\partial \bar{\chi}}{\partial r}\right)\frac{1}{r\bar{\rho}}\frac{\partial \bar{\psi}}{\partial z} - \frac{1}{r\bar{\rho}}\frac{\partial \bar{\psi}}{\partial r}\frac{\partial(C\bar{\chi})}{\partial z}\right] = g\frac{\partial}{\partial r}(\bar{\chi}^2 Q) +$$
$$\frac{\partial}{\partial z}(C\bar{\chi}^2 Q) - \frac{\partial}{\partial z}\left[\bar{\chi}\xi(F_\lambda + F_{sgs_\lambda})\right] + \left\{\frac{\partial}{\partial r}\left[\frac{1}{r\bar{\rho}}\frac{\partial \bar{\psi}}{\partial z}\frac{\partial(F_R\bar{\chi})}{\partial z}\right] + \frac{\partial}{\partial z}\left[F_R(-\frac{1}{r\bar{\rho}}\frac{\partial \bar{\psi}}{\partial z}\frac{\partial \bar{\chi}}{\partial r} + \frac{1}{r\bar{\rho}}\frac{\partial \bar{\psi}}{\partial r}\frac{\partial \bar{\chi}}{\partial z} + \bar{\chi}^2 Q) -$$
$$\bar{\chi}\frac{\partial F_R}{\partial t}\right]\right\}.$$

Similar to SEE, this is an elliptical partial differential equation but with additional radial eddy forcing terms. It can be solved using the same method of numerically solving SEE. Likewise, we can include different terms in Eq. (6) to derive SEE-like equations with different complexities, and use these diagnostic equations to evaluate how different factors regulate the overturning circulation in an unbalanced framework. We have been using this method to diagnose HWRF model output. The results will be reported in a separate paper.

We understand the reviewer's concern. Since Eq. (6) is not so critical for this study, we have removed it from the manuscript. The related sentences have been rewritten in the revised manuscript. Please see Page 5, lines 4 – 14 in the revised manuscript.

Continued:

Line 6: "In classic TC theories". Citations please.

Citations have been provided in the revised manuscript. The related sentences have been rewritten. Please see Page 5, line 4 in the revised manuscript.

Lines 7-8: This statement is incorrect. The eddy forcing terms can be zero, but acceleration terms may still be nonzero.

Lines 7-8? We couldn't find the sentence in the manuscript, but we agree on the comment that the eddy forcing can be zero, but acceleration may still be nonzero. This is easy to see from the azimuthal-mean tangential wind budget equation.

Line 10: Why is Equation (8) to be time differentiated? Please explain.

As we explained previously, the purpose to differentiate Eq. (8) with respect to time is to derive a diagnostic equation for the mean secondary overturning circulation. The paragraph has been rewritten in the revised manuscript.

10. Pg. 7, the text pertaining to Equations (9) and (10). My reading of this text is as follows: the HWRF model uses this two-component formulation for Km (i.e. the Hong and Pan closure in the BL/TL (Eq. 9) and the Smagorinsky closure with stability modification by Lilly above the BL/TL (Eq. 11), respectively) in the vertical and horizontal mixing terms for momentum, heat and moisture. Is this summary correct? Please clarify.

To answer this question and the question below, let's first review the main characteristics of turbulence. On the turbulent energy spectrum, large turbulent eddies are directly generated by the instabilities of the mean flow and obtain energy directly from the mean flow. These large energy-containing eddies, such as convective thermal plumes or boundary layer roll vortices generated by the inflection-point instability, are anisotropic. Large eddies then transport their energy through the energy cascade process to smaller-scale eddies. Smaller eddies contain less energy and are less flow-dependent and more isotropic than larger eddies. Eventually the eddy energy is dissipated into heat via molecular viscosity. On the energy spectrum, there is an intermediate range of scales known as inertial sub-range where the net incoming energy from larger-scale eddies is in equilibrium with the net energy cascading to smaller-scale eddies. The turbulent kinetic energy is neither generated nor dissipated in the inertial sub-range, just transferring from larger to smaller eddies. Eddies with scales smaller than inertial sub-range are commonly considered to be isotropic.

For numerical simulations with horizontal grid spacing smaller than inertial sub-range, such as large eddy simulations (LESs), large anisotropic energy-containing eddies are explicitly resolved by models, thus, only small eddies need to be parameterized.
Since eddies with scales smaller than inertial sub-range are isotropic, LES models treat the isotropic SGS horizontal and vertical turbulent mixing using 3D SGS turbulent mixing scheme, which is directly coded within the model dynamic core, rather than placing it a physics module outside the dynamic core. Smagorinsky SGS turbulent model (Smagorinsky 1965) is a 3D SGS mixing scheme widely used in LESs in which the eddy exchange coefficient is calculated by $K(x,y,z,t) = (c\Delta)^2 \left[ \frac{\partial \bar{u}_i}{\partial x_j} \left( \frac{\partial \bar{u}_i}{\partial x_j} + \frac{\partial \bar{u}_j}{\partial x_i} \right) \right]^{\frac{1}{2}}$, where $\Delta = (\Delta x \Delta y \Delta z)^{\frac{1}{3}}$ and $c$ is an empirical coefficient. Note that the original Smagorinsky scheme does consider the stability effect. In the later practice, many LES models use the Smagorinsky scheme along with stability correction, e.g. MacVean and Mason (1990) stability correction. Deardroff 3D TKE SGS scheme is another 3D SGS mixing scheme is widely used in LESs.

In contrast, for numerical models with large horizontal grid spacing (e.g., greater than 1 km), large energy-containing eddies are not resolved, but are part of the SGS processes. In this case, it is not appropriate to use 3D SGS scheme to parameterize these large eddies since they are anisotropic. Therefore, the vertical and horizontal SGS mixing must be treated differently

in these numerical models. The current method is to retain the mixing (or diffusion) model built within the dynamic core to treat the SGS horizontal mixing but to have a separate physics module often called PBL scheme (or more precisely the vertical turbulent mixing scheme) outside the dynamic core to handle SGS vertical mixing. The PBL scheme for treating vertical SGS mixing is a one-dimensional (1D) scheme.

Now for the reviewer's question, Eq. (9) and Eq. (10) in the manuscript only describe the vertical eddy exchange coefficients within and above the diagnosed PBL height used by the 1D PBL scheme module outside the dynamic core. The above-PBL eddy change coefficient, Eq. (10), is determined based on the resolved vertical shear and gradient Richardson number ($K_m = l^2 f_m(Ri_g)\sqrt{\left|\frac{\partial \tilde{u}}{\partial z}\right|^2 + \left|\frac{\partial \tilde{v}}{\partial z}\right|^2}$). This method has been adopted by many PBL schemes, such as YSU scheme. So, it is not the Smagorinsky diffusion model. As we stated previously, in numerical models, the Smagorinsky diffusion model is not a separate physics module but is built within the model dynamic core. In HWRF, the SGS diffusion model built within the dynamic core is a revised 2D Smagorinsky scheme, which is used to treat SGS horizontal mixing. The horizontal eddy exchange coefficient is determined by $K_h = L_h^2 D_h$, where $L_h$ is a tunable mixing length and $D_h = (\frac{\partial \tilde{v}}{\partial x} + \frac{\partial \tilde{u}}{\partial y})^2 + (\frac{\partial \tilde{u}}{\partial x} - \frac{\partial \tilde{v}}{\partial y})^2$ is the deformation (Zhang et al. 2018).

Bryan and Rotunno (2009) and Bryan (2012) use a much higher value of Km for horizontal diffusion than employed here. How do the authors explain this difference in model formulation compared to Bryan and Rotunno?

As explained previously, in mesoscale models, the SGS horizontal mixing and vertical mixing are treated separately because of the unresolved anisotropic large eddies. SGS horizontal mixing is handled within the model dynamic core by a built-in diffusion model, whereas SGS vertical mixing is treated by a separate physics module known as PBL scheme outside the dynamic core. Bryan and Rotunno (2009) and Bryan (2012) focused on the horizontal diffusion problem and investigated the sensitivity of TC evolution to horizontal eddy exchange coefficients by adjusting the tunable mixing length. In our study, we did not touch anything related to the SGS horizontal mixing. We only focused on the SGS vertical mixing, particularly, the vertical turbulent mixing above the PBL generated by the cloud processes associated with the eyewall and rainband convection.

This has been clarified in the revised manuscript. Please see Page 7, lines 10 – 19. Within the 1D PBL scheme framework, there are two ways that we may use to include in-cloud turbulent mixing parameterization in the scheme. The first method is what we did in this study to extend the diagnosed boundary layer so that the layer includes turbulence generated by both surface processes and cloud processes aloft. The disadvantage is that this method has problems to include the turbulence in the anvil clouds in the upper troposphere associated with outflow because the anvil clouds are detached from the PBL turbulence in a vertical model column. The second method is to keep the current 1D turbulent mixing parameterization framework unchanged but improve the stability calculation above the boundary layer by including cloud effects. Currently, most of the schemes used today

calculate the Brunt-Vaisala frequency as $N^2 = \frac{g}{\theta_0}\frac{\partial \overline{\theta_v}}{\partial z}$. But this is not sufficient to account for the buoyancy generated by clouds. Emanuel and Rotunno (2011) and Emanuel (2012) also suffered this problem when they calculated the Brunt-Vaisala frequency in the outflow. Using a parcel theory, Durran & Klemp (1982) showed that an accurate Brunt-Vaisala frequency in the saturated atmosphere should be expressed as $N_m^2 = g\left\{\frac{1+B}{1+A}\left[\frac{d\ln\theta}{dz} + \frac{1}{C_pT}\left(1 + \frac{q_s}{\varepsilon}\right)\left(AC_p\frac{dT}{dz} - Bg\right)\right] - \frac{dq_t}{dz}\right\}$, where $A = \frac{L^2 q_s}{C_p R_v T^2}$, $B = \frac{Lq_s}{RT}$, $q_s$ and $q_t$ are the saturated and total mixing ratio including condensate and precipitation. We have been testing this method in HWRF simulations (see our presentation at 2018 HFIP Annual Review Meeting). We will report the second method and the results in a separate paper.

Another advantage of the second method by calculating accurate Brunt-Vaisala frequency in clouds is that it may be used to improve horizontal SGS mixing parameterization. Current studies (e.g. Bryan and Rotunno 2009, Bryan 2012, Zhang et al. 2018) simply adjusted the mixing length, which does not have much physics behind it. In-cloud stability changes resulting from the accurate calculation of Brunt-Vaisala frequency in clouds can also affect horizontal SGS mixing. We shall investigate this issue in our future research.

11. Pg. 14, Lines 1-2. " . . . since the large energy-containing turbulent eddies are not resolved at the current model resolution of 2 km." What are these large energy-containing eddies in these simulations and in real-life tropical cyclones?

The unresolved large eddies at the model resolution of 2 km may include buoyancy-driven sub-kilometer convective cells or elements in the eyewall and rainbands and roll vortices generated by the inflection-point instability in the boundary layer. These large eddies can induce effective non-local vertical mixing. However, the above-PBL convective elements are difficult to parameterize since the stretching vertical grid used by models causes low vertical resolutions above the boundary layer. Even with the relative high vertical resolution within the boundary layer, Zhu (2008) showed that PBL schemes used in WRF cannot appropriately account for the vertical transport induced by roll vortices explicitly simulated by WRF-LES.

This has been clarified in the revised manuscript.

12. Figures 10 and 11. The resolved eddy-forcing tendency for the azimuthally averaged tangential velocity tendency equation is plotted in cross section and on several horizontal height surfaces. In these panels the units are displayed as inverse seconds. Shouldn't the units be that of acceleration (if instantaneous tendencies averaged over some finite time interval) or meters per second (if integrated over some time finite interval)? Please clarify here and elsewhere.

We are very sorry for this mistake. The unit of the tendency should be ms$^{-2}$. This has been corrected in the revised manuscript.

[revised manuscript text omitted]

---

## Referee Comment (RC2) · Anonymous Referee #2 · 6 Jul 2019

Review of

"Role of eyewall and rainband eddy forcing in tropical cyclone intensification." (acp-2018-610)

by Ping Zhu, Bryce Tyner, Jun A. Zhang, Eric Aligo, Sundararaman Gopalakrishnan, Frank D. Marks, Avichal Mehra, and Vijay Tallapragada

This study introduces an in-cloud turbulent mixing parameterization for the HWRF model. The rationale for this parameterization is that the classic HWRF PBL parameterization scheme does not account for intense mixing in eyewall/rainband clouds. The authors admit their scheme is a rather crude approximation of mixing, but it seems to help with producing better hurricane intensity predictions.

This is a promising study. Under the premise that the results are not cherry-picked, the improvements are quite astonishing. However, there are a number of issues that should be addressed to improve the manuscript. One of them is some amount of carelessness when describing the results and the figures. This and other issues are detailed below.

General Comments:

1. One of the weaknesses of this study is that the authors do not discuss why the eddy forcing would be responsible for TC spin-up. There are some hand-wavy arguments about interactions between the turbulence and microphysics but the reader is left in the dark with what's actually going on.

2. How does this work relate to the LES hurricane studies by George Bryan (or the LES work of the first author)? My recommendation is to relate this work to previous TC studies that employ an LES approach.

3. Show aggregate statistics of how much improvement the turbulence parameterization yielded. Even though the authors present more than just a case study, there is no mention of the results from all their simulations. If these aggregate results were included, there would be less suspicion about "cherry picking".

4. There is no discussion of how large the eddy exchange coefficient of momentum, $K_m$, should be (see also comment XX below). Can't you compute $K_m$ from your prior LES work and compare it to the values you get from the parameterization?

Specific Comments:

1. The title is misleading. Given this title, I'd expect a more quantitative study on the turbulent processes and their roles, but the actual manuscript is more about describing and applying the turbulence parameterization.

2. Page 6: "But cumulus schemes are not designed to account for the eddy forcing to the momentum, heat, and moisture budgets but rather serve as a means to remove the convective instability generated by the large-scale flow and alter the thermodynamic structure of the environment based on the parameterized convective fluxes and precipitation (Arakawa and Schubert 1974; Wu and Arakawa 2014)."
   —> This is not true for the CLUBB scheme
   (https://agupubs.onlinelibrary.wiley.com/doi/full/10.1002/2015GL063672).

3. Page 12: "In contrast, "TL-HWRF" produces a well-defined closed ring around the storm center that is clearly shown in both dynamic (vertical velocity, Fig. 8b) and thermodynamic (hydrometeor mixing ratio, Fig. 8d) fields."
   —> Actually, none of the panels in Fig. 8 show a closed ring (although the inner core is much more defined in the TL-HWRF runs). Furthermore, the comparison between observations and model (Figs. 7 and 8) is subjective, hand waving and and does not add anything of substance.

4. Page 13: "Comparing Fig. 11b with Fig. 10b, it is easy to see that the model-resolved eyewall eddy forcing above the PBL in the "TL-HWRF" experiment has a magnitude about 5 times larger than the corresponding SGS eddy forcing, suggesting that the resolved eddy processes provide a major forcing that drives the primary circulation of the TC vortex in this case."
   —> At first look this contradicts the overall statement that SGS turbulence is important. The authors should comment on this apparent contradiction.

5. Page 14: "other 4 major hurricanes" —> four other major hurricanes

6. Page 14: As another example, Figure 13 compares the satellite observed vortex inner-core structure of Harvey (2017) with the simulated ones by the two HWRFs during the early and middle stages of Harvey's RI. The asymmetric rainband structure, the closed ring feature around the storm center, and the size of the convective ring shown in satellite observations are reasonably reproduced by TL-HWRF."
   —>Subjective and hand wavy. For a better comparison, the panels should at least be plotted on the same lat/lon domain.

7. Page 14: "one may concern about" —> one may be concerned about

8. Fig. 3: Why is there no sign of surface friction?

9. Fig. 4e,f: I'm curious, why is there no indication of a melting layer in the reflectivity plots?

10. Fig. 5: Why are the $K_m$ values 2 and 5 km so much larger than at the surface? (this observation is based off the colorbar range, which goes from 0-80 in Fig. 5a, but from 0-300 or more in Figs. 5b, c).

---

## Author Comment (AC2) · 16 Aug 2019

This study introduces an in-cloud turbulent mixing parameterization for the HWRF model. The rationale for this parameterization is that the classic HWRF PBL parameterization scheme does not account for intense mixing in eyewall/rainband clouds. The authors admit their scheme is a rather crude approximation of mixing, but it seems to help with producing better hurricane intensity predictions.

This is a promising study. Under the premise that the results are not cherry-picked, the improvements are quite astonishing. However, there are a number of issues that should be addressed to improve the manuscript. One of them is some amount of carelessness when describing the results and the figures. This and other issues are detailed below.

**We very much appreciate the reviewer for his thoughtful and constructive comments. The manuscript has been revised accordingly based on the comments. We hope that the revised manuscript is satisfactory for publication. Below is the point-to-point response to reviewer's comments.**

General Comments:

1. One of the weaknesses of this study is that the authors do not discuss why the eddy forcing would be responsible for TC spin-up. There are some hand-wavy arguments about interactions between the turbulence and microphysics but the reader is left in the dark with what's actually going on.

**To date TC rapid intensification (RI) is still not fully understood. One of the arguments is that the processes governing RI are primarily axisymmetric. Vigh et al. (2018) showed that the observed RI of several major hurricanes can be well reproduced by the CHIPs model, a simple axisymmetric 2D radius-height dynamic model developed by Emanuel et al. (2004). Their results are consistent with previous studies by Nolan and Grasso (2003) and Nolan et al. (2007) who showed that it is the vortex axisymmetric response to the azimuthally-averaged diabatic heating (rather than the heating associated with individual asymmetries) that is responsible for the resultant intensity change in their simulations. These theoretical studies are supported by observations. From a large amount of 37 GHz microwave products, Kieper and Jiang (2012) showed that the appearance of a cyan color ring around the storm center is highly correlated to subsequent RI, provided that environmental conditions are favorable. This result is supported by the TRMM Precipitation Radar (PR) data (Jiang and Ramirez 2013; and Tao and Jiang 2015), which showed that nearly 90% of RI storms in different ocean basins formed a precipitation ring around the storm center prior to RI.**

**Back to the question of how eddy forcing contributes to the RI. The answer lies in the fact of how eddy forcing facilitates the realization of axisymmetric dynamics responsible for RI. A key difference between a 3D full physics simulation and a 2D axisymmetric simulation is that a TC vortex is prescribed to be axisymmetric in the latter, whereas in the former a simulated TC vortex is often not axisymmetric enough to efficiently convert latent heating to kinetic energy. Thus, whether a 3D full physics simulation can well capture RI depends on if TC internal dynamics including eddy forcing can organize the eyewall convection into a convective annulus. This may explain why the azimuthally averaged heating rate in a 3D full physics simulation is often smaller than that in a 2D axisymmetric simulation. However,**

there are examples in literature that showed the energy conversion in 3D full physics model simulations can be as efficient as that in an axisymmetric model simulation. For instance, Persing et al. (2013) reported that "there is a short period of time when the rate of spin-up in the 3D model exceeds that of the maximum spin-up rate in the axisymmetric model, and during this period the convection is locally more intense than in the axisymmetric model and the convection is organized in a quasi ring-like structure resembling a developing eyewall".

In our simulations, the modified HWRF is able to produce a well-defined quasi-closed ring around the storm center and the size of the ring is similar to the observed one, implying that the RI of simulated hurricanes is likely governed by the axisymmetric dynamics articulated by Vigh et al. (2018) and Nolan et al. (2007). In contrast, the default HWRF is unable to simulate the observed TC inner-core structure, suggesting that it fails to generate the TC axisymmetric dynamics needed for the RI. Our results also indicate that the SGS physics involving with the eyewall in-cloud turbulent mixing above the PBL and the resultant change in storm structure and resolved eddy forcing facilitate the realization of the axisymmetric dynamics responsible for RI in 3D full-physics simulations. Our simulations are consistent with Persing et al. (2013) who showed that the 3D eddy processes can assist the intensification process by contributing to the azimuthally averaged heating rate, to the radial contraction of the maximum tangential velocity, and to the vertical extension of tangential winds through the depth of the troposphere. The discussions/comparisons between our simulations and Persing et al. (2013) simulations have been included in the revised manuscript. Please see Page 2, line 23 – Page 3, line 7; Page 6, lines 13 – 25; Page 15, lines 18 - 25; Page 16, lines 17 – 23; Page 17, lines 17 – 20.

2. How does this work relate to the LES hurricane studies by George Bryan (or the LES work of the first author)? My recommendation is to relate this work to previous TC studies that employ an LES approach.

In numerical simulations, the high order terms caused by the nonlinearity of turbulent flow need to be parameterized in order to close the governing equations. In the state-of-the-art numerical models, a SGS parametric model is coded within the model dynamic core (or solver) to account for the SGS mixing. For convection permitting simulations with a horizontal grid-spacing greater than 1 km, large turbulent eddies with scales greater than Kolmogorov inertial subrange are not resolved. These energy-containing eddies generated directly by the instabilities of the mean flow are fundamentally anisotropic. A common method to account for the directional dependent turbulent transport induced by anisotropic eddies is to retain the SGS model built within the model dynamic solver to treat the horizontal turbulent mixing, but to have a separate physics module, often called the planetary boundary layer (PBL) scheme outside the dynamic solver to handle vertical turbulent mixing. The PBL scheme for treating vertical SGS mixing is a one-dimensional (1D) scheme. This method is used by HWRF and many other mesoscale models.

As model grid spacing reduces down to the inertial subrange, large energy-containing eddies are explicitly resolved, and thus, the only eddy processes that need to be parameterized are those with scales smaller than the inertial subrange. These small eddies are commonly considered to be isotropic. Because of this, there no need to have a separate module to treat

vertical turbulent mixing like convection permitting simulations. Rather, the horizontal and vertical SGS mixing induced by isotropic eddies can be handled by the same SGS model built within the model dynamic solver. This type of simulation is the so-called LES. The key is that it requires a 3D SGS model so that both the vertical and horizontal mixing induced by isotropic eddies can be appropriately parameterized. In this sense, the approach of our LES study is similar to that of Bryan et al. (2003), Green and Zhang (2015), and Zhu (2008) in terms of both model grid resolution and the way of treating SGS mixing.

Since eddies with scales smaller than inertial subrange contain much less energy and are less flow-dependent than large energy-containing eddies, the LES methodology is commonly thought to be insensitive to formulaic details and arbitrary parameters of the SGS model, and thus, the turbulent flow generated by LESs are often used as a proxy for reality and a basis for understanding turbulent flow and guiding theories when direct observations are difficult to obtain. In the past, LESs were mainly used to elucidate problems associated with the turbulent processes within the PBL. Here we use this approach to better understand the turbulent processes in the eyewall.

While using LES to simulate TC is promising, evaluation of the fidelity of the simulated TC vortex and the associated fine-scale structures resolved by LES is a challenge. In the absence of decisive observational measurements, the principal method of evaluating LES has been through sensitivity studies of individual LES models with different SGS mixing schemes or inter-comparisons among different LES models. The logic is that the robustness of the simulations testifies to its fidelity. Such sensitivity tests and inter-comparison studies in the past have shed favorable light on the LES approach in general in many meteorological applications, but they also raised questions about the ability of LES to realistically reproduce some unique features in the atmosphere. While there are individual LES studies of TCs, the sensitivity of LES to SGS parameterization has never been examined when the LES approach is used to simulate TCs. Such sensitivity tests are needed since intense turbulence in the eyewall can exist well beyond the PBL. Therefore, our LES work is motivated to gain insight into the global behavior of Giga-LES in TC modeling. We have tested three 3D SGS models commonly used in LESs: (a) 3D Smagorinsky SGS model (Smagorinsky, 1963), (b) 3D 1.5-order TKE SGS model (Deardorff, 1980), and (c) 3D nonlinear backscatter and anisotropy (NBA) SGS model (Kosović, 1997). This work has been completed and submitted to *Journal of Advances in Modeling Earth Systems*. Here we used some of the results from this study. This has been clarified in the revised manuscript. Please see Page 7, lines 9 – 18; Page 9, line 10 – Page 10, line 11.

3. Show aggregate statistics of how much improvement the turbulence parameterization yielded. Even though the authors present more than just a case study, there is no mention of the results from all their simulations. If these aggregate results were included, there would be less suspicion about "cherry picking".

We have been collaborating with the Environmental Modeling Center (EMC), NOAA, on this work. EMC tested the scheme in HWRF full cycle simulations using the HWRF operational version at that time. The results show that the modified HWRF significantly

**reduces the bias of maximum wind speed (Fig. 1). The total number of case simulations for different lead time is 1079. The results have been added in the revised manuscript. Please see Page 18, lines 13 – 17 and Figure 16.**

[Figure]

**Figure. 1: Maximum wind speed bias error (kt) as a function of forecast lead time (hr) averaged over all tested storms and cycles. Average bias errors are shown for the 2018 HWRF model baseline (H18C, blue), 2018 HWRF model including in-cloud turbulent mixing (H18P, cyan), and 2017 operational HWRF model (H217, red). The storms tested included Hermine (2016), Harvey (2017), Irma (2017), Maria (2017), and Ophelia (2017). The total number of cases for various forecast lead times is indicated by the cyan labels at the bottom (Courtesy to Dr. Sergio Abarca at EMC NOAA).**

4. There is no discussion of how large the eddy exchange coefficient of momentum, Km, should be (see also comment XX below). Can't you compute Km from your prior LES work and compare it to the values you get from the parameterization?

[revised manuscript text omitted]

Figure 2 and the related discussions have been included in the revised manuscript. Please see Page 10, line 12 – Page 11, line 21.

[Figure]

**Figure 2: (a): Azimuthal-mean radius-height distribution of the vertical momentum fluxes,** $\tau = (\overline{w'u'}^2 + \overline{w'v'}^2)^{\frac{1}{2}}$, **induced by the resolved eddies with scales smaller than 2 km from the WRF-LES that uses the 3D NBA SGS model. (b): Vertical profiles of the parameterized (dashed) and resolved (solid) vertical eddy exchange coefficients of momentum averaged over 30 – 60 km radii (where the eyewall is located) from the three LESs that use different 3D SGS models. Note that the results are averaged over 3 – 8 simulation hours and the SGS eddy exchange coefficients are the direct output from the 3D SGS models used in the simulations.**

Specific Comments:

1. The title is misleading. Given this title, I'd expect a more quantitative study on the turbulent processes and their roles, but the actual manuscript is more about describing and applying the turbulence parameterization.

Turbulent processes associated with TC eyewall and rainbands are complicated problems as they are beyond the conventional scope of the PBL. To our knowledge, the importance of eyewall turbulent mixing above the PBL to TC evolution has not been addressed before. In numerical simulations, the asymmetric eddy forcing to TC evolution consists of both resolved and SGS components. In this study, we showed that in the convection permitting simulations at grid resolution of 2 km, the resolved eddy forcing and the TC inner-core structure (and thus the TC intensity) are sensitive to the parameterization of the turbulent mixing in eyewall clouds above the PBL. Indeed, our manuscript describes the in-cloud turbulent mixing parameterization and its application in HWRF, but the main purpose of this paper is to demonstrate and explore the sensitivity of TC intensification to the change in eddy forcing resulting from the SGS turbulent mixing parameterization. As we stated in the summery of the manuscript, our treatment of in-cloud turbulent mixing itself is crude, and thus, the scheme may not be ready for use in operational TC forecasts in its current form. Nonetheless, our results show that numerical simulations of TC intensification are sensitive to the parameterization of SGS turbulent mixing induced by the cloud processes above the PBL in the eyewall and rainbands.

In fact, the PBL scheme used in HWRF is not the only scheme that has problems to represent in-cloud turbulent processes in deep convective clouds. Using WRF-ARW, we simulated a deep convective system with various PBL schemes available in WRF. These include the YSU (Hong et al. 2006), GFS (Hong and Pan 1996), ACM2 (Pleim 2007), MYJ (Janjic 1994), QNSE (Sukoriansky et al. 2005), MYNN-2.5/MYNN-30 (Nakanishi and Niino 2006, 2009), BouLac (Bougeault and Lacarrere 1989), UW (Bretherton and Sungsu 2009), and GB (Grenier and Bretherton 2001) schemes. None of them can appropriately generate the intense turbulent mixing in the deep convective clouds above the boundary layer. For TC simulations, we think that this problem is particularly important and should be addressed in future research. We agree that parts of our paper describes the method to treat in-cloud turbulent mixing and its application in HWRF, but because of this we are able to generate the appropriate eddy forcing above the PBL in the eyewall and rainbands, so that we can look into how eddy forcing in the eyewall and rainbands facilitates the axisymmetric dynamics underlying RI of TCs. We think that the title does reflects this part of our research. We admit that we could do more quantitative analyses on the turbulent processes and their roles in the TC intensification. But as we showed in the paper, the inclusion of an in-cloud turbulent mixing parameterization in the model induces changes not only in the resolved eddy forcing but also TC inner-core structure. On top of that, all these changes are entangled together, thus, we need to find an appropriate method to separate these changes. This will be the focus of our future research.

2. Page 6: "But cumulus schemes are not designed to account for the eddy forcing to the momentum, heat, and moisture budgets but rather serve as a means to remove the convective instability generated by the large-scale flow and alter the thermodynamic structure of the environment based on the parameterized convective fluxes and precipitation (Arakawa and Schubert 1974; Wu and Arakawa 2014)."
—> This is not true for the CLUBB scheme
 (https://agupubs.onlinelibrary.wiley.com/doi/full/10.1002/2015GL063672).

**We agree that our statement is not accurate. It is true that the original cumulus schemes developed by Arakawa and Schubert (1974) was to remove the convective instability generated by the large-scale flow and alter the thermodynamic structure of the environment based on the parameterized convective fluxes and precipitation. The turbulent mixing was not considered in this regard. However, some later developed more advanced cumulus schemes do consider the effects of turbulent mixing in schemes. The sentence has been rewritten in the revised manuscript. Please see Page 6, lines 1 – 8.**

3. Page 12: "In contrast, "TL-HWRF" produces a well-defined closed ring around the storm center that is clearly shown in both dynamic (vertical velocity, Fig. 8b) and thermodynamic (hydrometeor mixing ratio, Fig. 8d) fields."  —> Actually, none of the panels in Fig. 8 show a closed ring (although the inner core is much more defined in the TL-HWRF runs). Furthermore, the comparison between observations and model (Figs. 7 and 8) is subjective, hand waving and does not add anything of substance.

**We agree with the comments. The convective ring in "TL-HWRF" is not closed, but like the reviewer said, the eyewall structure in "TL-HWRF" is much more well defined than that in the default HWRF. Note that the figures do not means to provide a quantitative comparison between simulations and observations as an apple-to-apple comparison is not possible here because of the apparent difference in satellite images and simulations. However, they do provide a qualitative comparison of how different the TC inner-core structure is in the two HWRF simulations with and without an in-cloud turbulent mixing parameterization when comparing to observations. The inner-core structure difference between the two simulations is apparent and it shows how sensitive the TC inner-core structure to the parameterization of eyewall/rainband turbulent mixing above the PBL. This has been clarified in the revised manuscript and related sentences have been rewritten. Please see Page 14, lines 29 – 31; Page 15, lines 12 – 17.**

4. Page 13: "Comparing Fig. 11b with Fig. 10b, it is easy to see that the model resolved eyewall eddy forcing above the PBL in the "TL-HWRF" experiment has a magnitude about 5 times larger than the corresponding SGS eddy forcing, suggesting that the resolved eddy processes provide a major forcing that drives the primary circulation of the TC vortex in this case." —> At first look this contradicts the overall statement that SGS turbulence is important. The authors should comment on this apparent contradiction.

**This seemingly paradoxical result can be understood as follows. In a real TC, the eddy forcing consists of a continuous spectrum. But in numerical simulations, the eddy forcing, such as the one in Eq. 2, $\frac{D\bar{\bar{M}}}{Dt} = r(F_\lambda + F_{sgs\_\lambda})$, is split into the model-resolved and SGS components because of the discretized model grids. Although they appear as two separate terms in the governing equations and are determined separately in numerical simulations, the two split parts of eddy forcing are not independent but interact with each other. This means that we cannot simply judge the importance of resolved and SGS eddy forcings to TC intensification solely based on their individual magnitude, rather, we need to look at how the changes in the SGS parameterization induces the change in the resolved fields and vice versa. Such a mutual dependence is understandable since large energy-containing turbulent eddies, such as kilometer and sub-kilometer convective elements and roll vortices, are not resolved**

but parameterized at the current grid spacing of convection permitting simulations. As we showed in the paper, the inclusion of an in-cloud turbulent mixing parameterization substantially changes the structure of the simulated TC vortex, and thus, the changes in the resolved eddy forcing, as well as the axisymmetric TC dynamics. This has been clarified in the revised manuscript. Please see Page 4, lines 14 – 25; Page 20, lines 1 – 2.

5. Page 14: "other 4 major hurricanes" —> four other major hurricanes

**Corrected in the revised manuscript.**

6. Page 14: As another example, Figure 13 compares the satellite observed vortex inner-core structure of Harvey (2017) with the simulated ones by the two HWRFs during the early and middle stages of Harvey's RI. The asymmetric rainband structure, the closed ring feature around the storm center, and the size of the convective ring shown in satellite observations are reasonably reproduced by TLHWRF." —>Subjective and hand wavy. For a better comparison, the panels should at least be plotted on the same lat/lon domain.

**As we replied previously, the limitation of satellite images prevents us performing a quantitative comparison between simulations and observations. Figure 13 is meant to provide a qualitative view of how the inclusion of an in-cloud turbulent mixing parameterization alters the TC inner-core structure when comparing to the observed one. We take the suggestion; this figure has been replotted on the same lat/lon domain in the revised manuscript. Please see the updated Figure 14 in the revised manuscript.**

7. Page 14: "one may concern about" —> one may be concerned about

**Corrected in the revised manuscript.**

8. Fig. 3: Why is there no sign of surface friction?

**This is because the figure only shows the resolved vertical fluxes. It does not include the SGS fluxes.**

9. Fig. 4e,f: I'm curious, why is there no indication of a melting layer in the reflectivity plots?

**In the first version of the Ferrier-Aligo scheme implemented in 2014 version of HWRF, the scheme did include this bright band effect of melting layer. But in the later versions of HWRF, Ferrier and Aligo removed it because they felt that it was resulting in too broad of the convective region when looking at warm season MCSs. This is the reason why there is no indication of a melting layer in the reflectivity plot. But this will not affect our estimation of the depth of "TL" using radar reflectivity.**

10. Fig. 5: Why are the Km values 2 and 5 km so much larger than at the surface? (this observation is based off the colorbar range, which goes from 0-80 in Fig. 5a, but from 0-300 or more in Figs. 5b, c).

**As we showed previously, the calculated eddy exchange coefficients from LESs are also large above the PBL in the low-mid troposphere. This large eddy exchange coefficients likely result from the combined effects of large up-gradient vertical fluxes and small vertical gradient of mean variables in the eyewall.**

[revised manuscript text omitted]

---

## Author Response (AR2)

**Response to Editor**

Dear Editor,

Figure 16 has been replotted. The root mean square errors have been added in the figure. The figure caption has also been accordingly revised.

Thank you so much,

Ping

[revised manuscript text omitted]